**Data Availability Statement:** The data underlying the results presented in the study are available

# Analysis, identification and confirmation of synthetic opioids using chloroformate chemistry: Retrospective detection of fentanyl and acetylfentanyl in urine and plasma samples by EI-GC-MS and HR-LC-MS

Carlos A. Valdez[1,2,3]*, Roald N. Leif[1,2,3], Todd H. Corzett[1,3,4], Mark L. Dreyer[1,2,3]

**1** Forensic Science Center, Lawrence Livermore National Laboratory, Livermore, CA, United States of America, **2** Nuclear and Chemical Sciences Division, Lawrence Livermore National Laboratory, Livermore, CA, United States of America, **3** Physical and Life Sciences Directorate, Lawrence Livermore National Laboratory, Livermore, CA, United States of America, **4** Biosciences and Biotechnology Division, Lawrence Livermore National Laboratory, Livermore, CA, United States of America

* valdez11@llnl.gov

## Abstract

Electron Impact Gas Chromatography-Mass Spectrometry (EI-GC-MS) and High Resolution Liquid Chromatography-Mass Spectrometry (HR-LC-MS) have been used in the analysis of products arising from the trichloroethoxycarbonylation of fentanyl and acetylfentanyl in urine and plasma matrices. The method involves the initial extraction of both synthetic opioids separately from the matrices followed by detection of the unique products that arise from their reaction with 2,2,2-trichloroethoxycarbonyl chloride (Troc-Cl), namely Troc-norfentanyl and Troc-noracetylfentanyl. The optimized protocol was successfully evaluated for its efficacy at detecting these species formed from fentanyl and acetylfentanyl when present at low and high levels in urine (fentanyl: 5 and 10 ng/mL and acetylfentanyl: 20 and 100 ng/mL) and plasma (fentanyl: 10 and 20 ng/mL and acetylfentanyl: 50 and 200 ng/mL), values that reflect levels reported in overdose victims. The HR-LC-MS method's LOQ (limit of quantitation) for the Troc-norfentanyl and Troc-noracetylfentanyl products was determined to be ~10 ng/mL for both species. Even though the superiority in the detection of these species by HR-LC-MS over EI-GC-MS, the latter method proved to be important in the detection of the second product from the reaction, namely 2-phenylethyl chloride that is crucial in the determination of the original opioid. This observation highlights the importance of using complimentary analytical techniques in the analysis of a sample, whether biological or environmental in nature. The method herein serves as a complementary, qualitative confirmation for the presence of a fentanyl in collected urine, plasma and by extension other biological samples amenable to the common extraction procedures described for opioid analysis. More importantly, the method's main strength comes from its ability to react with unknown fentanyls to yield products that can be not only detected by EI-GC-MS and HR-LC-MS but can then be used to retrospectively identify an unknown fentanyl.

from the Lawrence Livermore National Laboratory, Forensic Science Center and are included in the Supporting Information accompanying this publication.

**Funding:** This work was performed under the auspices of the U. S. Department of Energy by Lawrence Livermore National Laboratory under Contract DE-AC52-07NA27344. The was funded fully by a Mid-Career Research Grant awarded by the Lawrence Livermore National Laboratory (PLS-21-FS-036) to C. A. V. The funders had no role in study design, data collection and analysis, decision to publish, or preparation of the manuscript.

**Competing interests:** The authors have declared that no competing interests exist.

## Introduction

Since its landmark synthesis by Paul Janssen [1, 2] and subsequent discovery of its effective, powerful, and safe anesthetic effects during perioperative surgical procedures [3] as well as for the management of pain in certain disease states [4, 5], fentanyl has become one of the most employed opioids in the field of medicine (Fig 1A). A few of the central attributes that make this synthetic opioid an unquestionable choice for physicians over morphine is its rapid onset time, powerful anesthetic profile (100 times more powerful than morphine) and the lower risk profile, relative to morphine, for acute heart and respiratory failure [6]. Unfortunately, the benefits of fentanyl use in the medical field has been overshadowed by its involvement in hundreds of thousands of overdose deaths due to its illicit use, along with other similar analogs such as acetylfentanyl (Fig 1A), outside the medical setting worldwide [7–9]. Furthermore, fentanyl analogs with more powerful anesthetic profile than the parent opioid, such as carfentanil and remifentanil, have been used as incapacitating agents [10]. Their status as an imminent threat to public health has been cemented by the countless cases worldwide and the mass production by clandestine laboratories using published protocols [11, 12]. As a direct counteroffensive move to this opioid crisis, numerous efforts by various government agencies and academic groups are being aimed at counteracting their toxic effects with drugs like naloxone and naltrexone (Fig 1B) [13–15], as well as developing more rapid ways for their early and efficient detection by Gas Chromatography-Mass Spectrometry (GC-MS) [16, 17] and Liquid Chromatography-Mass Spectrometry (LC-MS) [18, 19].

Collection of biological samples by medical personnel from deceased individuals who were exposed to fentanyl have included urine, plasma, blood (femoral and heart) and vitreous humor [16, 20]. Analysis for these opioids first involves their initial extraction using an organic solvent, followed by extract concentration and subsequent analysis by various analytical methods [16]. In most cases, fentanyl can be found in its intact form, but analysts may also encounter its most common metabolite, norfentanyl (Fig 1A) which is the result of the opioid's oxidative dealkylation by cytochrome P450 liver enzymes [21, 22]. Detection of fentanyl and analogs thereof has been accomplished using LC-MS and hyphenated methods like LC-MS/MS featuring detection limits for these substances down to the sub-nanogram level [23–26]. In

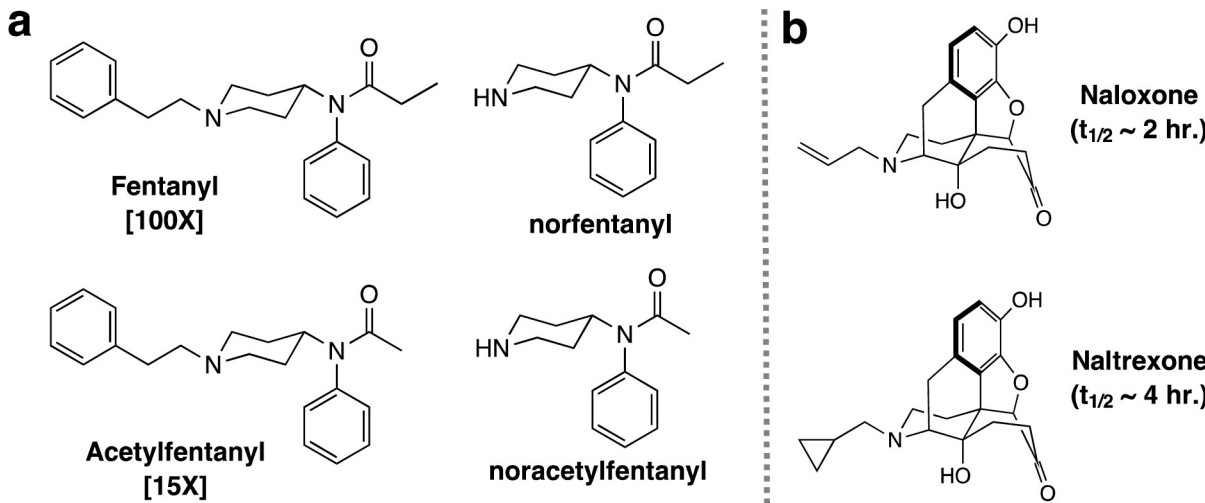

**Fig 1. Synthetic opioids belonging to the fentanyl family. (a)** Chemical structures of fentanyl, acetyfentanyl and their main metabolites, the potencies of each opioid relative to morphine are given in brackets; **(b)** commonly employed antidotes against fentanyl poisoning along with their circulation half-lives.

contrast, GC-MS has not enjoyed that level of success for a couple of reasons. The first one is the fact that the technique is inherently less sensitive than LC-MS and the second one is that fentanyls when encountered in biological samples are in their salt form thus requiring the use of additional neutralization/extraction steps that often results in analyte loss. Having the option of analyzing a sample, be it from a biological or environmental collection, by either LC-MS, GC-MS or both, to identify these synthetic opioids remains an invaluable tool in the fields of analytical chemistry and the emerging one of chemical forensics [27, 28]. Even though GC-MS methods possess some degree of limitations when compared to LC-MS methods, the technique can still be used as a complementary analytical method, as most of GC-MS instrumentation are equipped with mass spectral libraries for thousands of compounds in collections like the NIST and OCAD libraries, that greatly aid in the initial identification of an unknown material during the early stages of sample analysis [29–31].

Recently, our laboratory introduced a method for the indirect analysis of fentanyl and analogs based on their chemical modification with 2,2,2-trichloroethoxycarbonyl chloride (Troc-Cl) [32]. The reaction between fentanyl and more complex members of this family of synthetic opioids resulted in the formation of two products, that aside from being easily identified by EI-GC-MS, can be pieced together to retrospectively identify the original fentanyl (Fig 2A). We demonstrated that the method worked efficiently in a panel of ten representative fentanyls showing its broad-spectrum quality with LLOD and LLOQ values of 10 and 20 ng/mL respectively for fentanyl [32]. Interestingly, reaction of norfentanyl with Troc-Cl affords the same product that forms from the reaction of fentanyl itself with the chloroformate. Thus, the method herein not only results in the complete modification of any circulating, unmetabolized fentanyl to yield Troc-norfentanyl, but reaction of norfentanyl itself with Troc-Cl causes a further accumulation of this signature product in the victim's tissues that can be traced back to the original opioid (Fig 2B). In this report, we describe the proficiency of the method in the analysis of fentanyl and acetylfentanyl separately in human plasma and urine samples containing opioid levels comparable to those found in overdose victims for each (0.2–43 ng/mL for fentanyl in plasma and 2.2–450 ng/mL in urine and 153–260 ng/mL for acetylfentanyl in

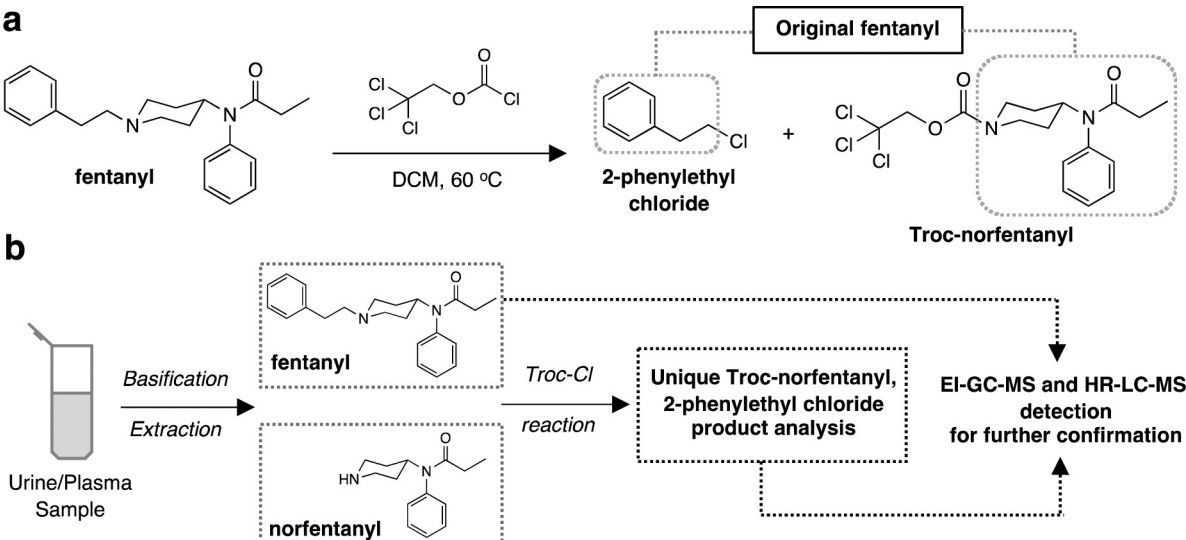

**Fig 2. Chemical modification of fentanyls. (a)** Reaction of fentanyl with 2,2,2-trichloroethoxycarbonyl chloride (Troc-Cl) to yield two unique, abiotic products that bear structural features of the original opioid (grayed boxes); **(b)** overall protocol to aid in the confirmation of fentanyl and norfentanyl in a biological sample by EI-GC-MS and targeted HR-LC-MS.

plasma and 230–240 ng/mL in urine) [33–35]. The method's overall description is presented in Fig 2B and entails an initial extraction step of the opioids from the biological matrix. After all the extracts are collected and evaporated, they are reacted with Troc-Cl to give the two unique, abiotic products Troc-norfentanyl (in the case of fentanyl) and 2-phenylethyl chloride. Therefore, the analysis of the initial extract would provide a first look at the fentanyl in question, while the reaction with Troc-Cl would provide a confirmatory identification of the fentanyl by analysis of the two predictable products arising from the reaction.

## Materials and methods

### Chemicals and reagents

All chemicals were purchased from commercial suppliers and used as received. 2,2,2-Trichloroethoxycarbonyl chloride (Troc-Cl), ammonium hydroxide and dichloromethane were purchased from Sigma-Aldrich (St. Louis, MO.). Potassium bicarbonate was purchased from Acros Organics (Westchester, PA.). Autosampler vials were purchased from Agilent Technologies (Santa Clara, CA.). Fentanyl and acetylfentanyl were synthesized using published protocols [11]. Thin layer chromatography (TLC) was used to monitor the production of amine-containing intermediates leading to the synthesis of fentanyl and acetylfentanyl using Merck 60-F254 sheets and detection accomplished with UV light ($\lambda$ = 254 nm) in conjunction with CAM [36, 37] and iodine vapor [38, 39]. All fentanyls as well as the Troc-norfentanyl and Troc-noracetylfentanyl standards were purified by flash column chromatography using a Biotage Isolera purification system.

### EI-GC-MS analysis method

A 6890 Agilent GC with 5975 MS detector equipped with a split/splitless injector was used for the analysis as previously described [40–43]. The GC column used for the analysis was an Agilent HP- 5ms UI capillary column (30 m × 0.25 mm id × 0.25 μm film thickness). Ultra-high purity helium, at 0.8 mL/min, served as the carrier gas. The inlet was operated in pulsed splitless mode (25 psi for 1 minute, followed by a 50 mL/min purge flow), with the injector temperature set at 250˚C and the injection volume was 1 μL. The oven temperature program was as follows: 40˚C, held for 3 min, increased at 8˚C/min to 300˚C, held for 3 min. The MS ion source and quadrupole temperatures were 230˚C and 150˚C, respectively. Electron impact (EI) was used with an ionization energy of 70 eV. The MS was operated to scan from $m/z$ = 29 to 600 in 0.4 sec with a solvent delay of 3.5 min.

### HR-LC-MS analysis method

A Thermo Scientific Vanquish Flex HPLC with a Thermo Scientific Q Exactive HF-X was used for the analyses. The LC column used for the analysis was a Waters Acquity HSS T3, 1.8 μm. Ultra-high purity nitrogen served as the collision gas. Mobile phases used were A = Water/0.1% formic acid and B = Acetonitrile/0.1% formic acid. The solvent gradient was 25 min long: Initial 1%, hold for 2 minutes then ramp to 10% B over 6 minutes. Ramp to 95% B over 7 minutes, hold at 95% B for 2 minutes and re-equilibrate at 1% B for 8 minutes. Samples (10 μL) were injected into the LC-HRMS for analysis. The oven temperature was 40˚C and the MS was operated to scan from m/z = 75 to 750 with a solvent delay of 3.5 min. MS acquisition was performed on a Thermo Scientific Q Exactive HF-X mass spectrometer operated using heated electrospray ionization (HESI) in positive ion mode was used with high resolution accurate mass to $\leq$ 3 ppm. The MS experiment is composed of a full MS spectrum (m/z = 75–750) at a resolving power setting of 30,000 (Full width at half maximum (FWHM) at

m/z = 200) followed by Data Dependent MS/MS with normalized higher-energy collisional dissociation (HCD) energies of 30% at resolving power of 30,000 (FWHM at m/z = 200).

## Sample preparation and extraction

Pooled human plasma and urine were purchased from Innovative Research (Novi, MI.) and were spiked with fentanyl and acetylfentanyl based on concentrations reported in these tissues from overdose victims [17]. Thus, fentanyl was spiked in urine (high = 10 ng/mL; low = 5 ng/mL) and plasma (high = 20 ng/mL; low = 10 ng/mL) while acetylfentanyl was spiked in urine (high = 100 ng/mL; low = 20 ng/mL) and plasma (high = 200 ng/mL; low = 50 ng/mL). Liquid-liquid extraction of the spiked matrices was performed using 1-chlorobutane and following established protocols for their extraction in biological matrices [44]. Specifically, urine or plasma (1 mL) was combined with 1.32 M ammonium hydroxide (pH 12, 2 mL) and vortexed. The urine and plasma samples were extracted using 4 mL and 12 mL of 1-chlorobutane respectively. After vortexing at 3,000 rpm for 30 seconds, the samples were centrifuged at 5,000*g* for 5 minutes to separate organic and aqueous layers, with the organic (top) layer being transferred into a clean vial. The organic layer was then dried using a nitrogen stream to give a residue that was then taken up in DCM (1 mL), treated with Troc-Cl (100 μL), potassium bicarbonate (5 mg) and heated to 60˚C for 3 hours. After 3 hours, the vial was cooled to ambient temperature. An aliquot of the reaction mixture (100 μL) was transferred to another autosampler vial equipped with a glass insert for GC analysis (injection volume: 1 μL).

## Results and discussion

Methods for the analysis of fentanyls by GC-MS abound, and many of these involve protocols that have been optimized for highest extraction efficiency such as Liquid-Liquid [45] and Dispersed Liquid-Liquid Micro-Extraction [46] followed by analysis. Other methods involve extensive work in their analytical part, with some reducing the analysis run time down to six minutes [47] and while others feature longer times, they provide the analyst with an effective way of studying multiple opioids in one single run as highlighted by the Direct Analysis in Real Time Mass Spectrometry (DART-MS) and Direct Sample Analysis Mass Spectrometry (DSA-MS) approaches [48, 49]. Irrespective of this, GC-MS has not been the analytical technique of choice when the analysis of ultra-low levels of fentanyl and related opioids are involved with only few examples achieving the low detection limits (pg/mL) [50, 51] that LC-MS has featured over the years. This lack of sensitivity by the technique can be attributed to several factors. One is that fentanyls whether encountered in a biological matrix or in an environmental setting (*e.g.*, drug confiscation) are in their salt form (*e.g.*, hydrochloride) that requires a basic sample preparation step for GC detection [52, 53]. Another factor is the inability to derivatize fentanyls using traditional derivatizing agents for converting them into species that can be detected by GC-MS. Intrigued by the challenges posed by fentanyl's chemical inertness, we explored the use of chloroformate chemistry to investigate its potential for modifying this opioid. To this end, we found that reaction between fentanyls and Troc-Cl yields two unique and predictable products bearing structural features from the original fentanyl [32]. In our original report, we demonstrated the extent of this reaction using only DCM as the organic medium. Now, using the optimized parameters for the reaction, we moved on to applying the feasibility of using this approach in the analysis of biological matrices under more relevant conditions. The two chosen matrices for our studies were plasma and urine which are two of the most often collected biological fluids from an overdose victim during autopsy for subsequent drug screening aside from femoral blood and vitreous humor [20].

The extraction of the spiked matrices was performed following a well-established method for their extraction from plasma [44]. After the extraction, the organic layer was evaporated to dryness, taken up in DCM, treated with Troc-Cl and $KHCO_3$ and then analyzed by EI-GC-MS. Our first set of experiments involved the analysis of the fentanyl-spiked plasma samples (Fig 3). For these experiments, plasma samples were spiked at two concentrations, a low (10 ng/mL) and a high (20 ng/mL), in order to explore the limits for the protocol's efficiency within the concentration window in overdose victims (~3.1–4.3 ng/mL) [54]. As a first step, the spiked samples were extracted and after their concentration, analyzed for fentanyl. As expected these extractions exhibited a complex GC chromatogram (Fig 3A). Analysis of the GC trace by single ion extraction (m/z = 245, base peak) for fentanyl was needed to detect the opioid and demonstrated that the initial extraction step worked well for both, the low and the high concentration spiked samples (Fig 3B and 3c). When the extracted residues were treated with an excess of Troc-Cl in the presence of potassium bicarbonate in DCM (Fig 3D), it was found that the reaction marginally performed for the high fentanyl concentration sample, while not providing a clear detection of the Troc-norfentanyl adduct for the low concentration sample (Fig 3E and 3F). In an effort to cleanly locate the Troc-norfentanyl product the single ion extraction (m/z = 149, base peak) mode was attempted (Fig 3E and 3F). As it can be appreciated from Fig 3E and 3F, the single ion extraction approach yielded a peak for only the high concentration sample while providing no discernible peak for the low concentration one. Furthermore, closer inspection of the peak in the chromatogram obtained in Fig 3F that appears using the single ion mode extraction (rt = 33.03 min.) possesses a more complex mass spectrum than anticipated for the Troc-norfentanyl. Upon careful analysis it becomes evident that this complex spectrum belongs to more than one species, one of which is the desired Troc-norfentanyl (Fig 4A). Key ions found in this spectrum that arise from Troc-norfentanyl include its base peak m/z = 149 and the almost equally intense and diagnostic m/z = 150 peak [32]. However, it is only these large ion peaks that can be gleaned *a priori* by studying the spectrum in Fig 4A while an interesting observation can be made in the area where the molecular ion peak for Troc-norfentanyl should appear (m/z = 406). An enlargement of this area is shown in Fig 4B, and one can observe the peaks in grey arising from another Troc-modified species that does not only exhibit an unfortunately close ion peak to Troc-norfentanyl's molecular ion peak but also features the same retention time. In Fig 4B, the molecular ion peak components for Troc-norfentanyl are outlined in red and due to the low abundance of this material in the mixture, it is difficult for it to overtake the larger signal arising from the other Troc-containing interference. Nevertheless, by marking all the peaks arising from Troc-norfentanyl (indicated with red arrows in Fig 4A) and adding the analysis of the molecular ion peak it can be confidently stated that the Troc-norfentanyl is present in the mixture. Comparison of the mass spectra for all components in this initial set of experiments shows that clearer and cleaner mass spectra are obtained for the high concentration samples over the low concentration ones, and this is a direct result from the opioid concentration difference between the two spiking levels (Fig 4C and 4D). One specific example in this case is the presence of several species in the GC chromatograms for the analysis of 2-phenylethyl chloride (insets in Fig 4E and 4F) where the product in the high concentration sample yields a fairly clear peak relative to the low concentration sample where it appears among other, equally abundant components. Lastly, an interesting point that arises from the collected set of four mass spectra is that it highlights the use of this protocol in retrospectively identifying the original fentanyl opioid (Fig 4A, 4B and 4F) if the concentration is high enough (20 ng/mL), recall that this analysis was not possible for the low fentanyl spiked plasma sample (10 ng/mL) where only the 2-phenylethyl chloride was identified unambiguously by GC-MS (Fig 4E). In the event where fentanyl was not the originally spiked opioid, one can anticipate that if 2-phenylethyl chloride is not observed in

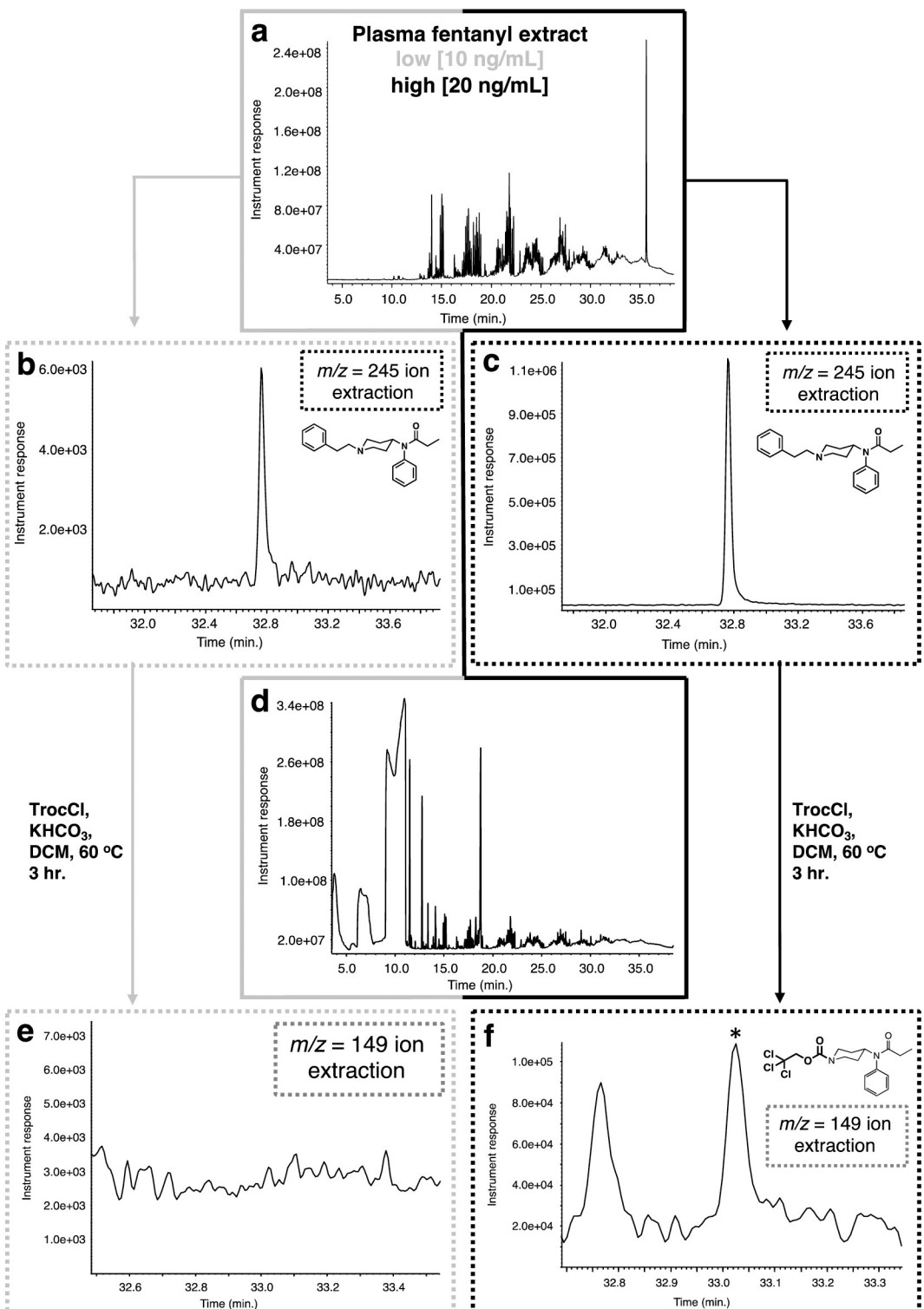

**Fig 3. Extraction and reaction of fentanyl with Troc-Cl when spiked in plasma at two concentrations (10 and 20 ng/mL).**
**(a)** GC chromatogram of extracted plasma sample; **(b)** extracted ion (m/z = 245) chromatogram for fentanyl (rt = 32.8 min.)
for low concentration spike; **(c)** extracted ion (m/z = 245) chromatogram for fentanyl for high concentration spike; **(d)** GC
chromatogram of the reaction between extracted plasma sample and Troc-Cl; **(e)** extracted ion (m/z = 149) chromatogram
for Troc-norfentanyl for the low concentration sample showing no discernible peak while **(f)** extracted ion (m/z = 149)
chromatogram for Troc-norfentanyl for the high concentration sample shows a clear peak (rt = 33.03 min.) composed of
Troc-norfentanyl.

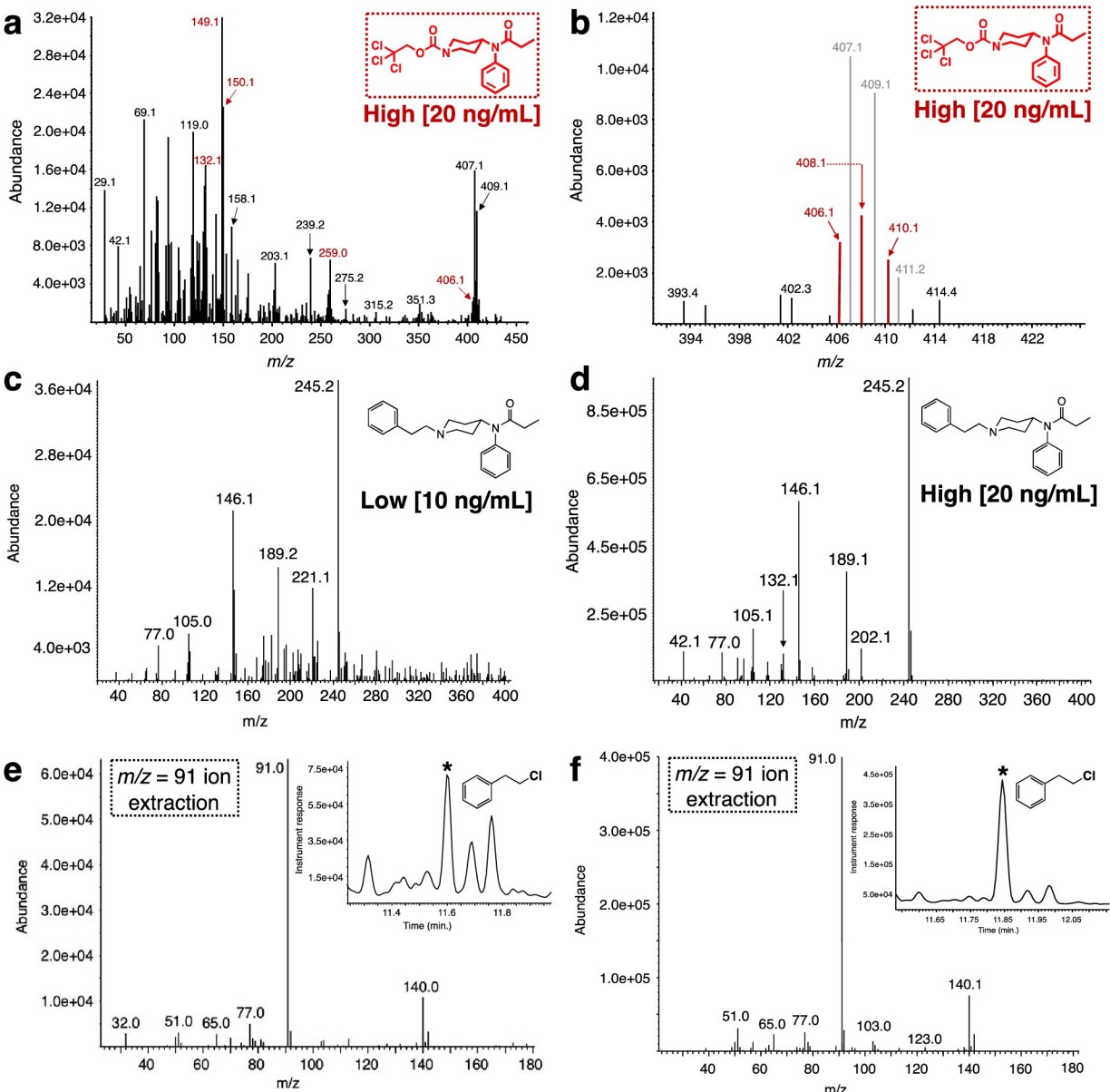

**Fig 4. Mass spectral data associated with the reaction between fentanyl-spiked plasma matrices with Troc-Cl.** Mass spectra for **(a)** Troc-norfentanyl product arising from the treatment of fentanyl-spiked plasma extract (20 ng/mL) with Troc-Cl, note that the spectrum is complex and it is the result of an additional Troc-containing interference (Troc-norfentanyl peaks indicated in red); **(b)** Expansion of the molecular ion peak region for Troc-norfentanyl where the peaks arising from our product are highlighted in red and those from the Troc-containing interference are in gray; mass spectra for **(c)** extracted fentanyl when spiked at 10 ng/mL and **(d)** extracted fentanyl when spiked at 20 ng/mL; **(e)** mass spectrum for 2-phenylethyl chloride for the lowest fentanyl concentration spike and **(f)** for the highest fentanyl concentration.

conjunction with Troc-norfentanyl, then those two pieces of information potentially point towards the existence of another fentanyl analog (*e.g.*, benzylfentanyl).

Analysis of the fentanyl-spiked urine samples provided similar results (Fig 5). The urine samples were spiked with fentanyl, similarly to the plasma samples, to yield two final concentrations, a low (5 ng/mL) and a high one (10 ng/mL). These samples were extracted as described above to provide a residue after concentration that exhibits an expected, complex GC chromatogram (Fig 5A). Analysis of the GC chromatogram using single ion extraction

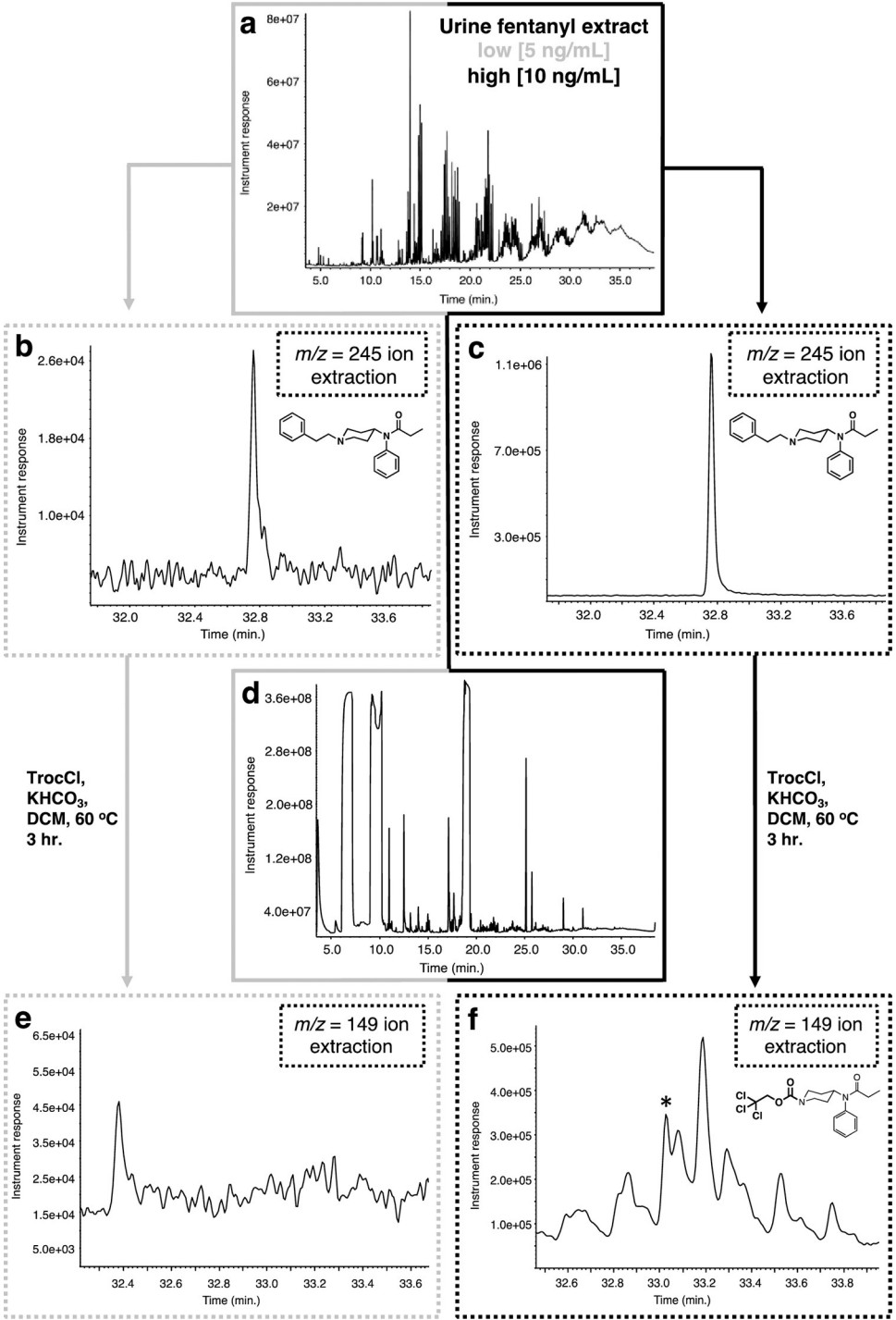

**Fig 5. Extraction and reaction of fentanyl with Troc-Cl when spiked in urine. (a)** GC chromatogram of extracted urine sample; **(b)** extracted ion (m/z = 245) chromatogram for fentanyl (rt = 32.8 min.) for low concentration urine spike (5 ng/mL); **(c)** extracted ion (m/z = 245) chromatogram for fentanyl for high concentration urine spike (10 ng/mL); **(d)** GC chromatogram of the reaction between extracted urine sample and Troc-Cl; **(e)** extracted ion (m/z = 149) chromatogram for Troc-norfentanyl for the low concentration urine sample (rt = 33.03 min.); **(f)** extracted ion (m/z = 149) chromatogram for Troc-norfentanyl for the high concentration spiked urine sample.

mode (m/z = 245, base peak) for fentanyl demonstrated that the extraction procedure was successful for both spiked samples (Fig 5B and 5C). When both residues (low and high) were reacted with an excess of Troc-Cl in the presence of potassium bicarbonate in DCM (Fig 5D) it was found that the reaction failed to provide the main product Troc-norfentanyl (Fig 5E) for the low concentration sample, while it performed poorly for the high concentration sample (asterisk in Fig 5F). Analysis for the Troc-norfentanyl product in both samples was accomplished using single ion extraction mode (m/z = 149, base peak). Again the discussion will be centered on the high concentration sample as no Troc-norfentanyl was detected for the low concentration sample. As shown previously for the fentanyl-spiked plasma samples (Fig 4A and 4B), extraction of the Troc-norfentanyl mass spectrum using the m/z = 149 base peak yields a similar spectrum to the one previously observed along with the other impurity that has undergone the chemical modification with Troc-Cl. A diagnostic peak in the Troc-norfentanyl spectrum aside from the m/z = 149 is the highly abundant m/z = 150 (Fig 6C). Once more, enlargement of the area in the mass spectrum where the molecular ion should occur shows that peaks belonging to Troc-norfentanyl's molecular ion peak, with its distinct chlorine isotopic signature can be clearly detected confirming the existence of this product from the reaction (Fig 6D). The peaks highlighted in red in Fig 6D belong to Troc-norfentanyl while the gray ones (shifted by one unit) belong to an unknown compound that has also reacted with Troc-Cl and possesses the trichloroethoxycarbonyl tag as witnessed by the similar chlorine isotopic signature to that one of Troc-norfentanyl. In Fig 6D, the molecular ion peak components for Troc-norfentanyl are outlined in red and unfortunately due to the low abundance of this material in the mixture, it is difficult for it to overtake the larger signal arising from the other Troc-containing interference. It appears that this concentration seems to establish a limit for the protocol in efficiently demonstrating the presence of fentanyl in a given sample. Detection of the other product, 2-phenylethyl chloride, also provided some difficulties at this concentration (10 ng/mL). The mass spectrum of 2-phenylethyl chloride after its ion extraction using m/z = 91 yielded a peak that clearly is the contains an underlying impurity (Fig 6E and 6F). For example a large peak (m/z = 120) is present in the spectrum for 2-phenylethyl chloride in addition to the ones belonging to this material (Fig 6F). Even though the analysis and detection of fentanyl using this methodology when dealing with very low concentrations (5 ng/mL) is difficult and not unambiguous, one must bear in mind that these concentrations are the lowest encountered in overdose situations, while in most cases the opioids are in much higher concentrations typically over 50 ng/mL. In addition, this approach is powerful when is used in conjunction with another method of analysis such as HR-LC-MS that can detect these species at these low concentrations without any issues. Furthermore, as it will be explained in the second part of this work where the HR-LC-MS results are disclosed, detection of 2-phenylethyl chloride is no achievable by LC-MS thus highlighting the use of GC-MS and LC-MS in complementary fashion.

After evaluating the protocol's efficiency with fentanyl, we turned our attention to acetylfentanyl. which is a lower molecular weight analog of fentanyl. Acetylfentanyl is 15x more powerful than morphine and due to its ease of production, it has become involved in many fatal overdose cases in the United States [55–57] and other parts of the world [58]. In similar fashion to the fentanyl experiments described above, plasma samples were spiked with acetylfentanyl at two concentrations, a low one (50 ng/mL) and a high (200 ng/mL). The spiked plasma samples were extracted as described above to provide a residue after concentration that again exhibits a complex GC chromatogram (Fig 7A). Analysis of the GC trace using the single ion extraction mode (m/z = 231, base peak) for acetylfentanyl demonstrated that the extraction procedure worked extremely well for both, the low and the high concentration spiked samples (Fig 7B and 7C). When both residues (low and high) underwent the Troc-Cl reaction (Fig 7D)

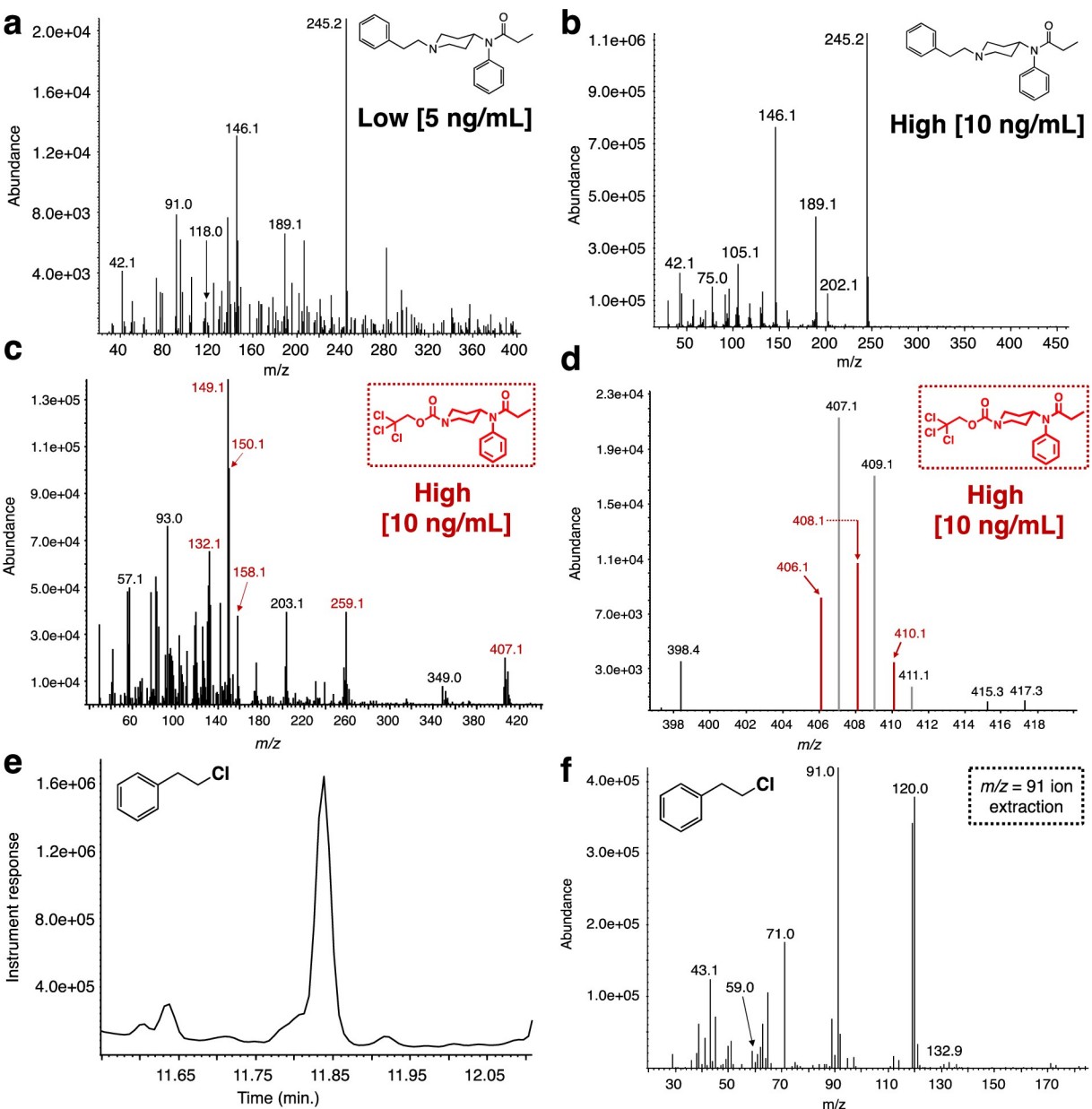

**Fig 6. Mass spectral data associated with the reaction between fentanyl-spiked urine matrices with Troc-Cl.** Mass spectra for **(a)** extracted fentanyl when spiked at 5 ng/mL; **(b)** extracted fentanyl when spiked at 10 ng/mL; **(c)** mass spectrum of Troc-norfentanyl product arising from the treatment of fentanyl-spiked urine extract (10 ng/mL) with Troc-Cl; **(d)** expansion of molecular ion peak for Troc-norfentanyl; **(e)** 2-phenylethyl chloride, the second confirmatory by-product from the Troc-Cl reaction; **(f)** mass spectrum of 2-phenylethyl chloride containing interfering signals from other matrix components.

it was found that the reaction successfully worked for both set of samples (Fig 7E and 7F), again using the single ion extraction (m/z = 135) mode for one of the products, Troc-norace-tylfentanyl. Comparison of the mass spectra for all components for these experiments demonstrates the ability of the protocol to correctly find and identify both products arising from the reaction of Troc-Cl and acetylfentanyl (Fig 8A–8F). In contrast to the fentanyl-containing plasma samples, which were spiked at a tenfold lower concentration than these, the clear mass spectra belonging to Troc-noracetylfentanyl (Fig 8B and 8E) can be appreciated. Thus, aside

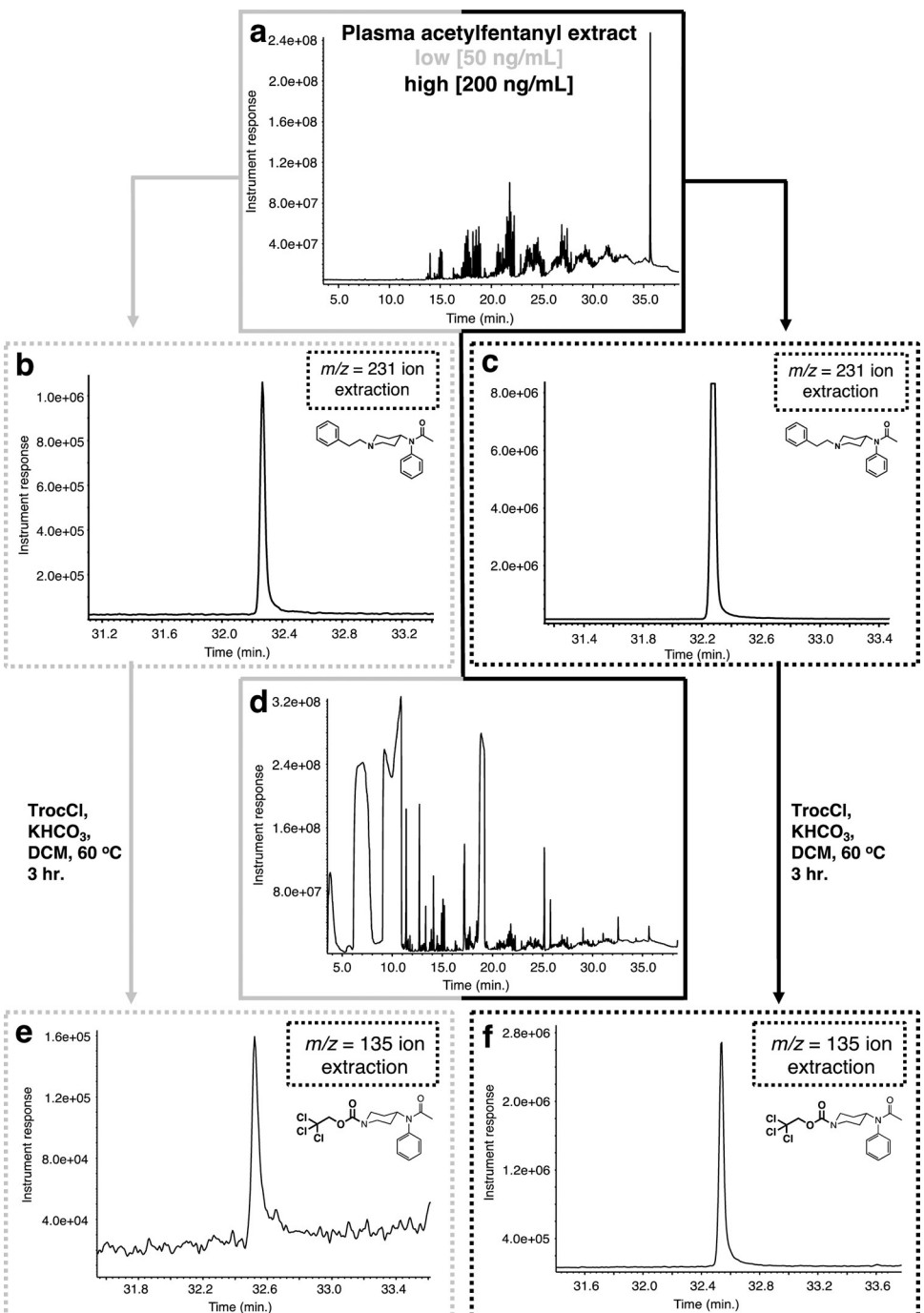

**Fig 7. Extraction and reaction of acetylfentanyl with Troc-Cl when spiked in plasma.** (a) GC chromatogram of extracted plasma sample; (b) extracted ion (m/z = 231) chromatogram for acetylfentanyl (rt = 32.3 min.) for low concentration plasma spike (50 ng/mL); (c) extracted ion (m/z = 231) chromatogram for fentanyl for high concentration plasma spike (200 ng/mL); (d) GC chromatogram of the reaction between extracted plasma sample and Troc-Cl; (e) extracted ion (m/z = 135) chromatogram for Troc-norfentanyl for the low concentration plasma sample (rt = 32.6 min.); (f) extracted ion (m/z = 135) chromatogram for Troc-norfentanyl for the high concentration spiked plasma sample.

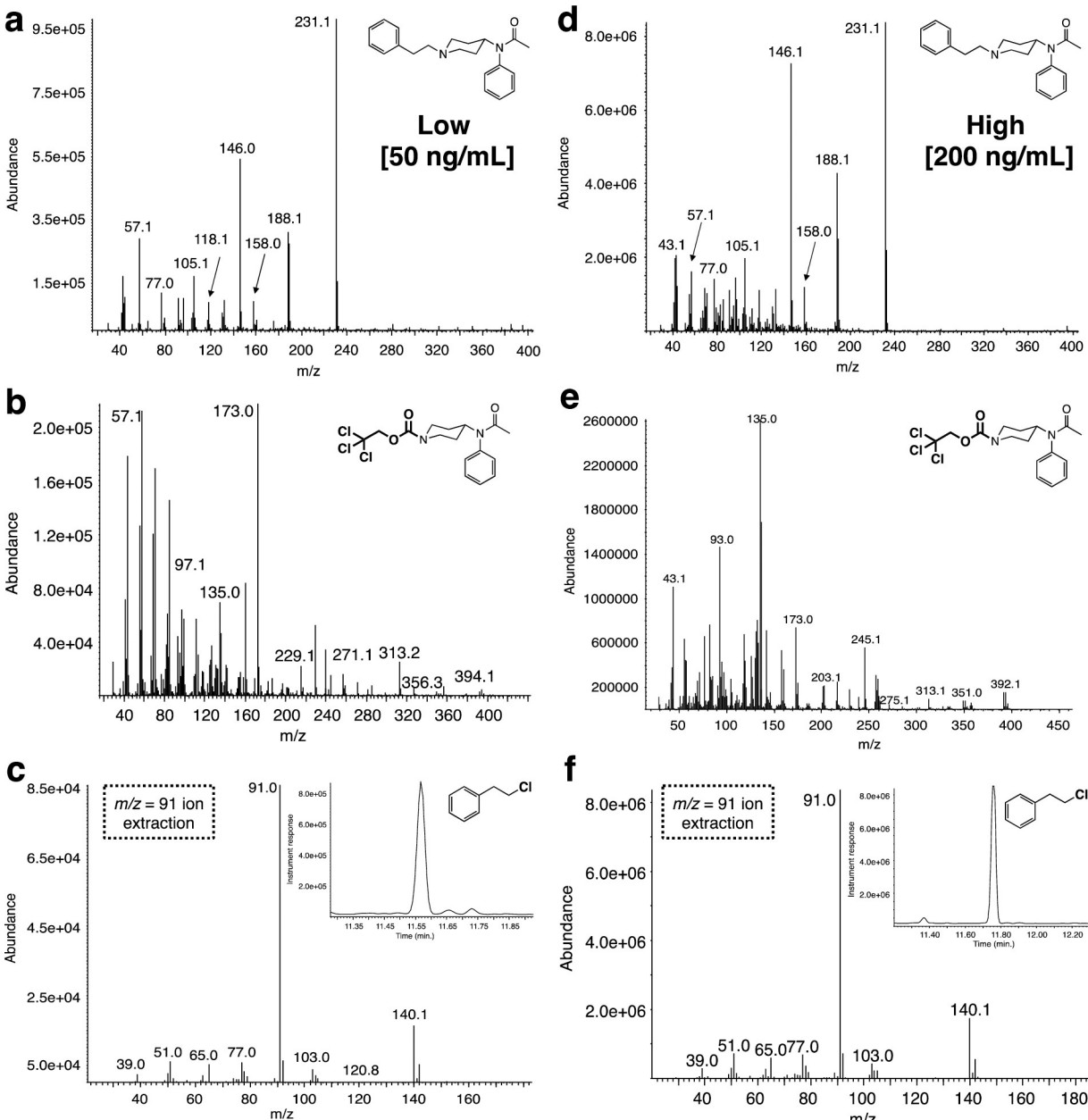

**Fig 8. Mass spectral data associated with the reaction between acetylfentanyl-spiked plasma matrices with Troc-Cl.** Mass spectra for **(a)** extracted fentanyl when spiked at 50 ng/mL; **(b)** Troc-norfentanyl product arising from the treatment of fentanyl-spiked plasma extract (50 ng/mL) with Troc-Cl; **(c)** and 2-phenylethyl chloride, the second confirmatory by-product from the reaction; mass spectra for **(d)** extracted fentanyl when spiked at 100 ng/mL; **(e)** Troc-norfentanyl product arising from the treatment of fentanyl-spiked plasma extract (100 ng/mL) with Troc-Cl; **(f)** and 2-phenylethyl chloride for the highest concentration spike.

from the base peak for this material (m/z = 135), other peaks that are in much lower abundance showing the typical chlorine-containing isotopic signatures can be observed (m/z = 392, 394).

The urine samples were spiked with acetylfentanyl free base at two concentrations, a low (20 ng/mL) and a high (100 ng/mL). The spiked samples were extracted as described above to provide a residue with a complex GC chromatogram (Fig 9A). Again, analysis of the GC trace

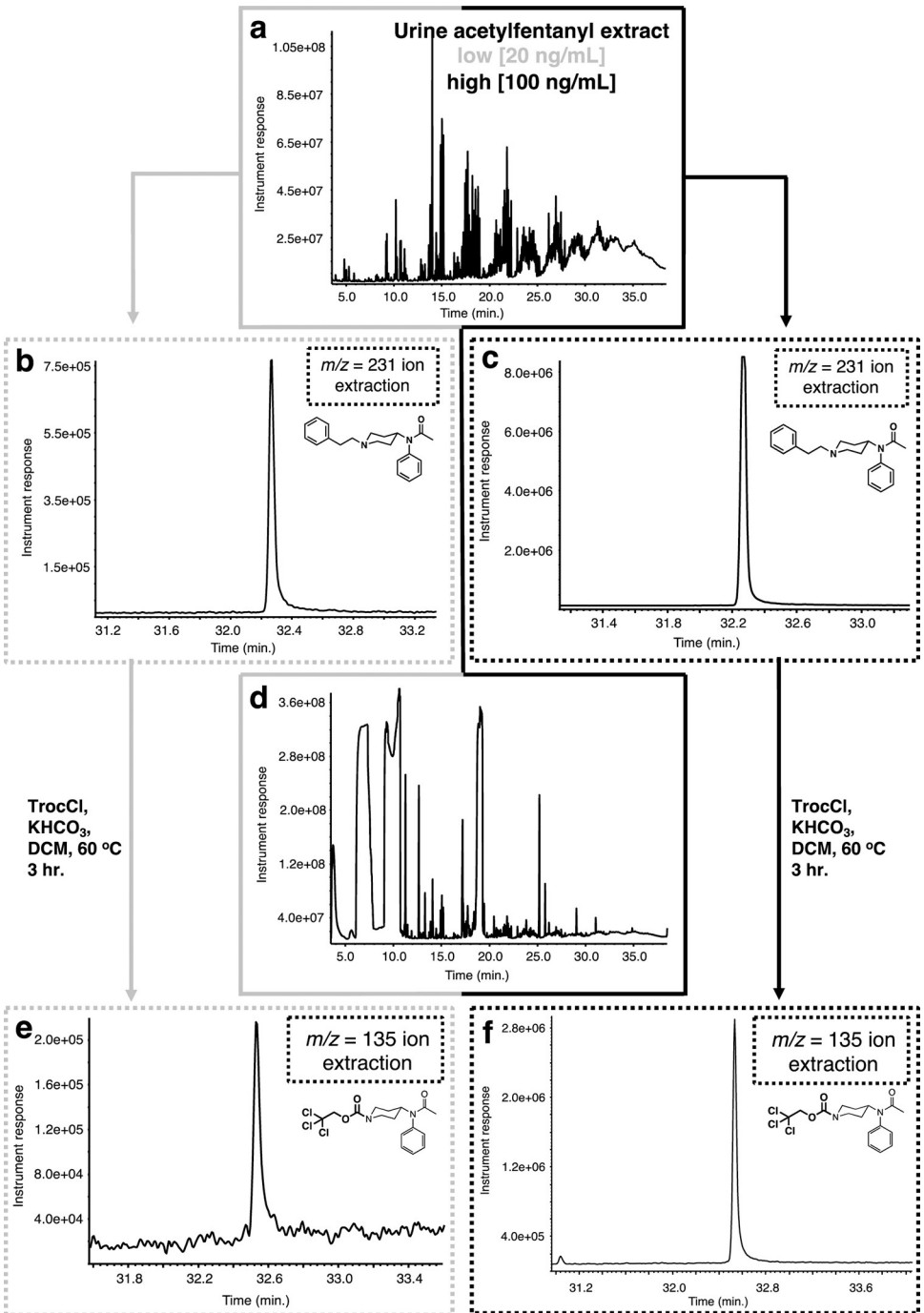

**Fig 9. Extraction and reaction of acetylfentanyl with Troc-Cl when spiked in urine. (a)** GC chromatogram of extracted urine sample; **(b)** extracted ion (m/z = 231) chromatogram for acetylfentanyl (rt = 32.3 min.) for low concentration urine spike (20 ng/mL); **(c)** extracted ion (m/z = 231) chromatogram for fentanyl for high concentration urine spike (100 ng/mL); **(d)** GC chromatogram of the reaction between extracted urine sample and Troc-Cl; **(e)** extracted ion (m/z = 135) chromatogram for Troc-norfentanyl for the low concentration urine sample (rt = 32.6 min.); **(f)** extracted ion (m/z = 135) chromatogram for Troc-norfentanyl for the high concentration spiked urine sample.

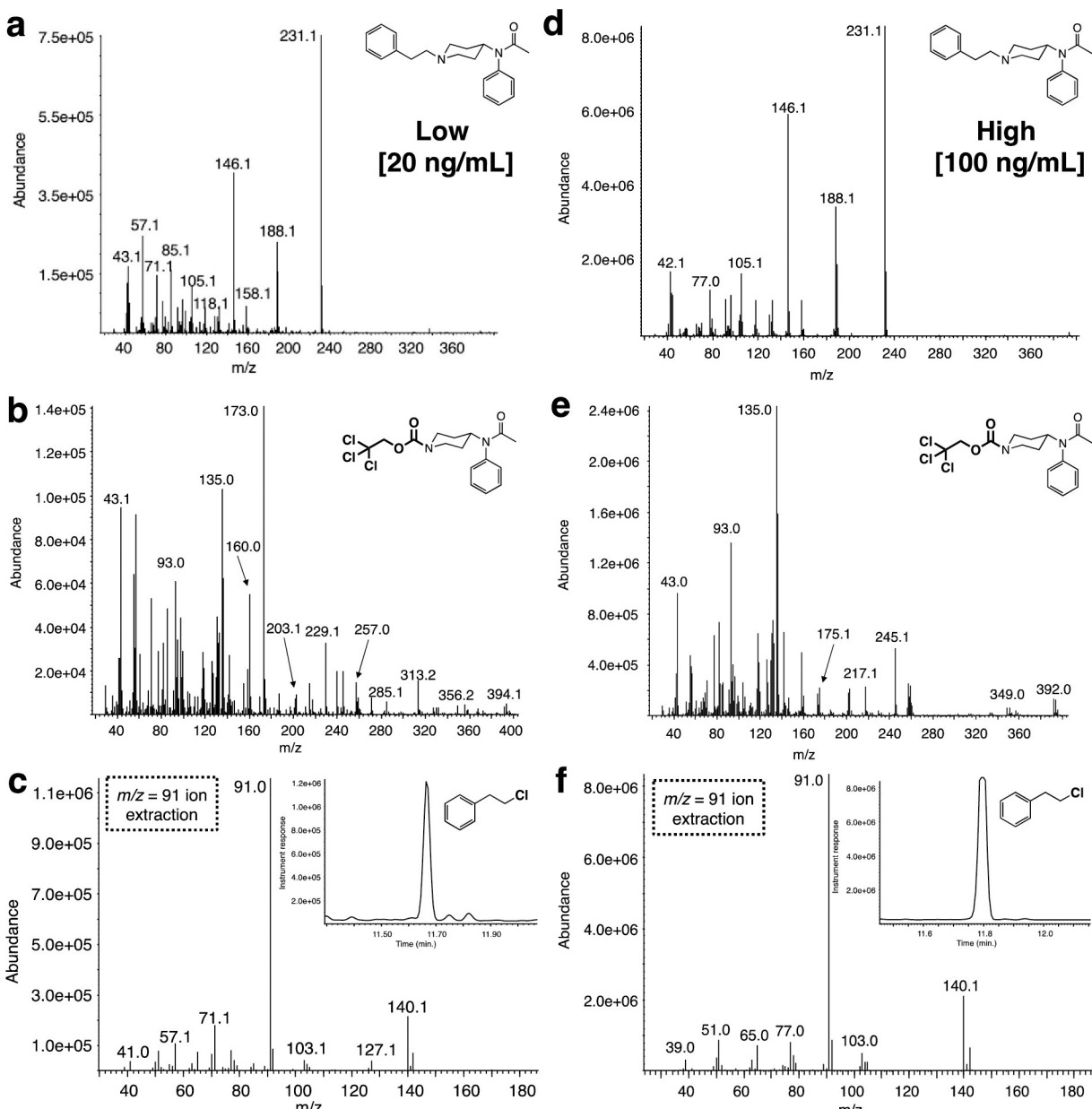

**Fig 10. Mass spectral data associated with the reaction between acetylfentanyl-spiked urine matrices with Troc-Cl.** Mass spectra for **(a)** extracted fentanyl when spiked at 20 ng/mL; **(b)** Troc-norfentanyl product arising from the treatment of fentanyl-spiked urine extract (20 ng/mL) with Troc-Cl; **(c)** and 2-phenylethyl chloride, the second confirmatory by-product from the reaction; mass spectra for **(d)** extracted fentanyl when spiked at 100 ng/mL; **(e)** Troc-norfentanyl product arising from the treatment of fentanyl-spiked plasma extract (100 ng/mL) with Troc-Cl; **(f)** and 2-phenylethyl chloride for the highest concentration spike.

using the single ion extraction mode (m/z = 231, base peak) for acetylfentanyl demonstrated that the extraction procedure worked well for both sets (Fig 9B and 9C). When both residues were reacted with Troc-Cl (Fig 9D) it was found that the reaction yielded the desired products (Fig 9E and 9F). Comparison of the mass spectra for all components in these set of experiments shows again that clearer and cleaner mass spectra are obtained for the high concentration samples over the low ones (Fig 10A and 10D). Again, an interesting point that arises from the collected set of four mass spectra comprising the low and high samples for the Troc-

noracetylfentanyl and 2-phenylethyl chloride, is that it highlights the use of this protocol in retrospectively identifying the original fentanyl (Fig 10B–10F).

In general, fentanyl analysis by LC-MS is greatly aided by two key features in the molecule, one is the presence of UV chromophores (an isolated phenyl ring and a phenylamido moiety) that significantly help in the overall detection of the fentanyl using a UV detector. Furthermore, and additional physical characteristic is the tertiary piperidine nitrogen that protonates in the acidic media commonly used for LC-MS analysis increasing the drugs' solubility in the aqueous gradients used. Analysis of the extracted plasma samples yielded complex total ion chromatograms (TICs) on which additional analytical interrogation can be undertaken (Fig 11A). Gratifyingly, identification of fentanyl on these extracts from both sets of samples (low and high) proved that the extraction part of the protocol was successful. Detection of the fentanyl involved extraction of its high-resolution mass for the opioid in its protonated form using the molecular formula ($C_{22}H_{29}N_2O^+$ [M+H$^+$ = 337.2274]) using a ± 5 ppm window. Extraction of this mass from the TIC using this approach yielded the ion chromatograms shown in Fig 11B and 11C. Reaction of the residue with Troc-Cl provides a complex mixture of substances that can be appreciated in Fig 11D. As it has been observed in with this matrix during their GC-MS analysis, that some natural plasma components may undergo the same chemical modification of the opioid. In similar fashion to the analysis of fentanyl extracts, high resolution mass extraction of these mixtures (for $C_{17}H_{22}Cl_3N_2O_3^+$ [M+H$^+$ = 407.0691] ± 5 ppm window) led to the direct detection of Troc-norfentanyl for both low and high concentration samples (Fig 11E and 11F). At this point it is noteworthy to mention the importance of analyzing samples by additional and complementary techniques like EI-GC-MS and HR-LC-MS. In contrast, EI-GC-MS analysis of the Troc-Cl treated mixtures specifically with the low concentration spiked samples yielded no detectable signals for Troc-norfentanyl and marginally for the second by-product of the reaction, 2-phenylethyl chloride. Therefore, in this case HR-LC-MS would seem to be the only technique to employ as its level of detection is orders of magnitude superior to the one featured by EI-GC-MS. This limitation of the technique for the detection of both component that can then be pieced together to figure out the structure of the original fentanyl, highlights the importance of having an additional technique like GC-MS to act as a complementary analytical tool. Regarding the fentanyl-spiked urine samples, these were extracted prior to their modification with Troc-Cl, and then analyzed for intact fentanyl as described above for the plasma samples. As expected, the extracts exhibited a complex LC chromatogram as appreciated by the TIC shown in Fig 12A. Analysis of the LC traces derived from the low and high concentration samples by using the same high resolution mass extraction approach for the protonated fentanyl form ($C_{22}H_{29}N_2O^+$ [M+H$^+$ = 337.2274] ± 5 ppm window) yields clear ion chromatograms for the presence of the opioid (Fig 12B and 12C). Again, these results proved that the extraction part of the protocol was successful and set the stage for the subsequent reaction with Troc-Cl. Reaction of both residues (low and high concentrations) with Troc-Cl provides a complex mixture of substances as appreciated in the TIC in Fig 12D. As it has been observed in with this matrix during their GC-MS analysis, some urine components may undergo the same chemical modification of the opioid adding substantial complexity to the TIC. In similar fashion to the analysis of fentanyl extracts, high resolution mass extraction of these mixtures for the Troc-norfentanyl product ($C_{17}H_{22}Cl_3N_2O_3^+$ [M+H$^+$ = 407.0691] ± 5 ppm window) led to the direct detection of Troc-norfentanyl for both, the low and high concentration samples (Fig 12E and 12F).

After evaluating the protocol's efficiency with fentanyl, we turned our attention to acetylfentanyl. One striking difference that becomes quickly apparent is that concentrations levels for acetylfentanyl used relative to fentanyl above are much higher and this translates into the quality of data that follows (Figs 13 and 14). The first set of experiments involved the analysis

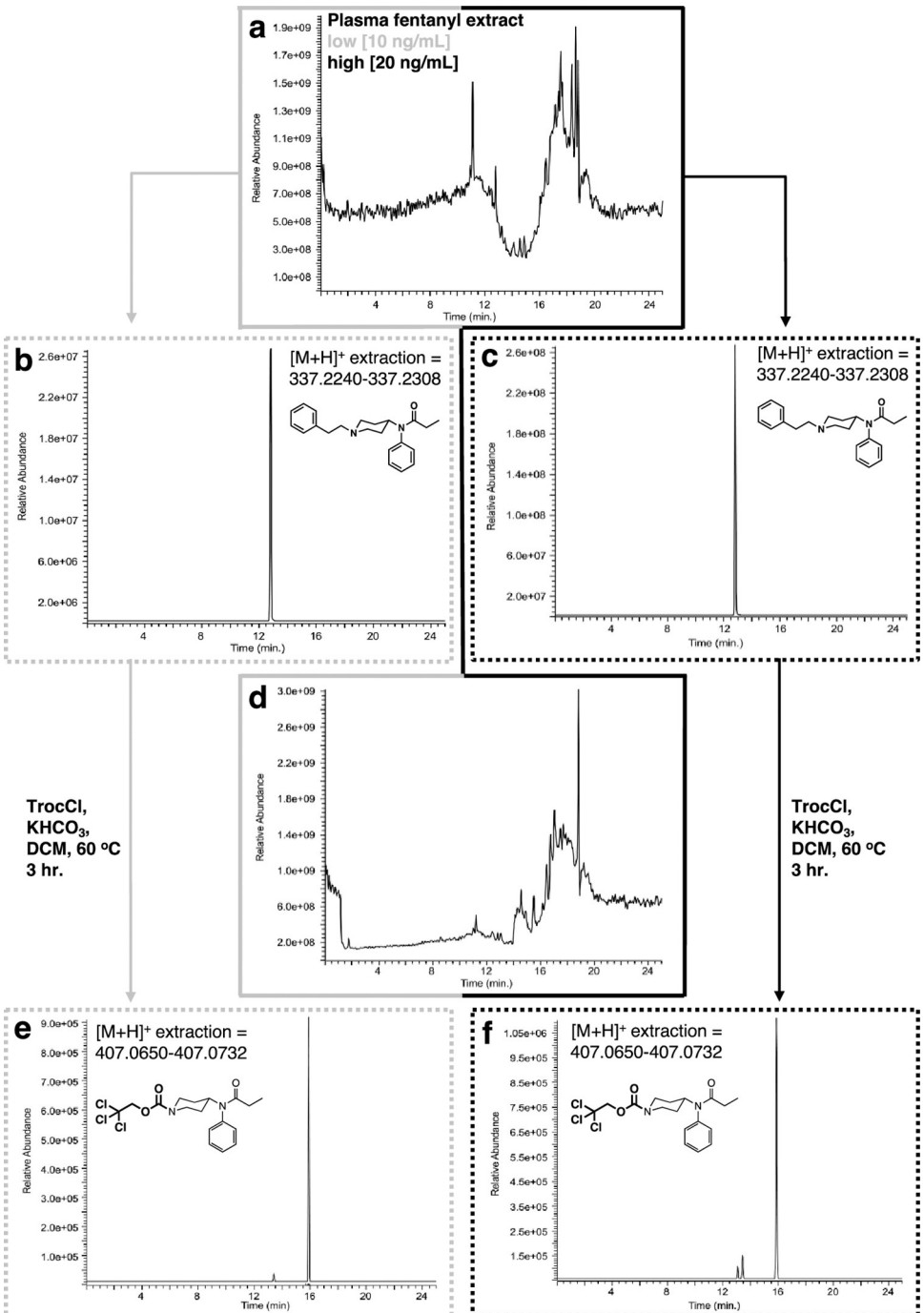

**Fig 11. Analysis by HR-LC-MS of extraction and reaction of fentanyl with Troc-Cl when spiked in plasma. (a)** Total ion chromatogram (TIC) of extracted plasma sample; **(b)** extracted ion ([M+H$^+$] = 337.2274 ± 5 ppm) for fentanyl (rt = 13.85 min.) for low concentration plasma sample (10 ng/mL); **(c)** extracted ion ([M+H$^+$] = 337.2274 ± 5 ppm) for high concentration in plasma sample (20 ng/mL); **(d)** Total ion chromatogram (TIC) of reaction between extracted plasma sample with Troc-Cl; **(e)** extracted ion ([M+H$^+$] = 407.0691 ± 5 ppm) for Troc-norfentanyl for low concentration spiked plasma sample; **(f)** extracted ion ([M+H$^+$] = 407.0691 ± 5 ppm) for Troc-norfentanyl for the high concentration spiked plasma sample showing its appearance at rt = 15.91 min.

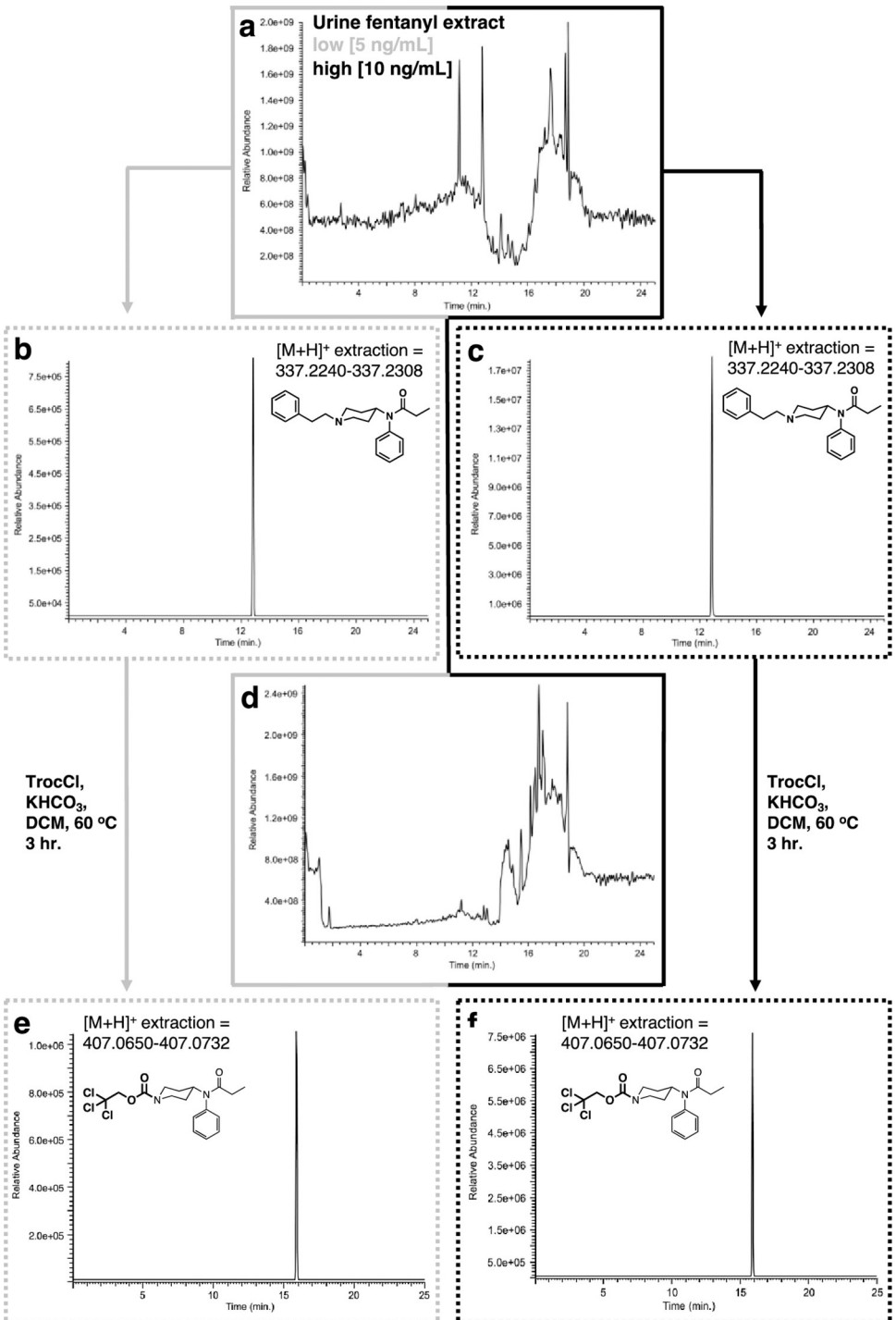

**Fig 12. Analysis by HR-LC-MS of extraction and reaction of fentanyl with Troc-Cl when spiked in urine. (a)** Ion chromatogram of extracted urine sample; **(b)** extracted ion ([M+H$^+$] = 337.2274 ± 5 ppm) for fentanyl (rt = 13.85 min.) for low concentration urine sample (5 ng/mL); **(c)** extracted ion ([M+H$^+$] = 337.2274 ± 5 ppm) for high concentration in urine sample (10 ng/mL); **(d)** Ion chromatogram of reaction between extracted urine sample with Troc-Cl; **(e)** extracted ion ([M+H$^+$] = 407.0691 ± 5 ppm) for Troc-norfentanyl for low concentration spiked urine sample; **(f)** extracted ion ([M+H$^+$] = 407.0691 ± 5 ppm) for Troc-norfentanyl for the high concentration spiked urine sample showing its appearance at rt = 15.91 min.

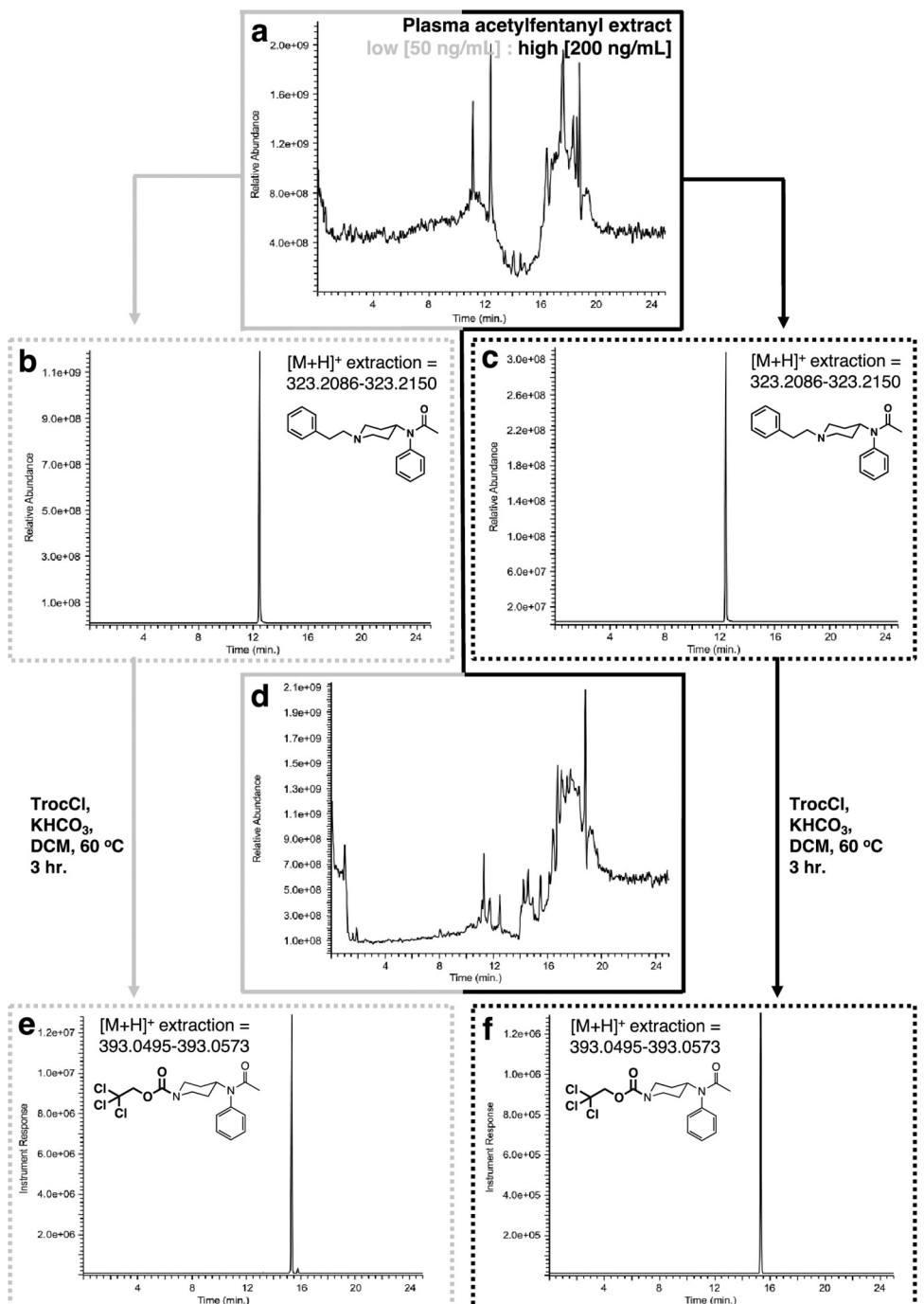

**Fig 13. Analysis by HR-LC-MS of extraction and reaction of acetylfentanyl with Troc-Cl when spiked in plasma.** (**a**) Total ion chromatogram (TIC) of extracted plasma sample; (**b**) extracted ion ([M+H]$^+$ = 323.2118 ± 5 ppm) for acetylfentanyl (rt = 12.46 min.) for low concentration plasma sample (50 ng/mL); (**c**) extracted ion ([M+H]$^+$ = 323.2118 ± 5 ppm) for acetylfentanyl for high concentration plasma sample (200 ng/mL); (**d**) Total ion chromatogram (TIC) of the reaction of extracted urine sample with Troc-Cl; (**e**) extracted ion ([M+H$^+$] = 393.0534 ± 5 ppm) for Troc-norfentanyl for low concentration spiked plasma sample; (**f**) extracted ion ([M+H$^+$] = 393.0534 ± 5 ppm) for Troc-norfentanyl for the high concentration spiked plasma sample showing its appearance at rt = 15.42 min.

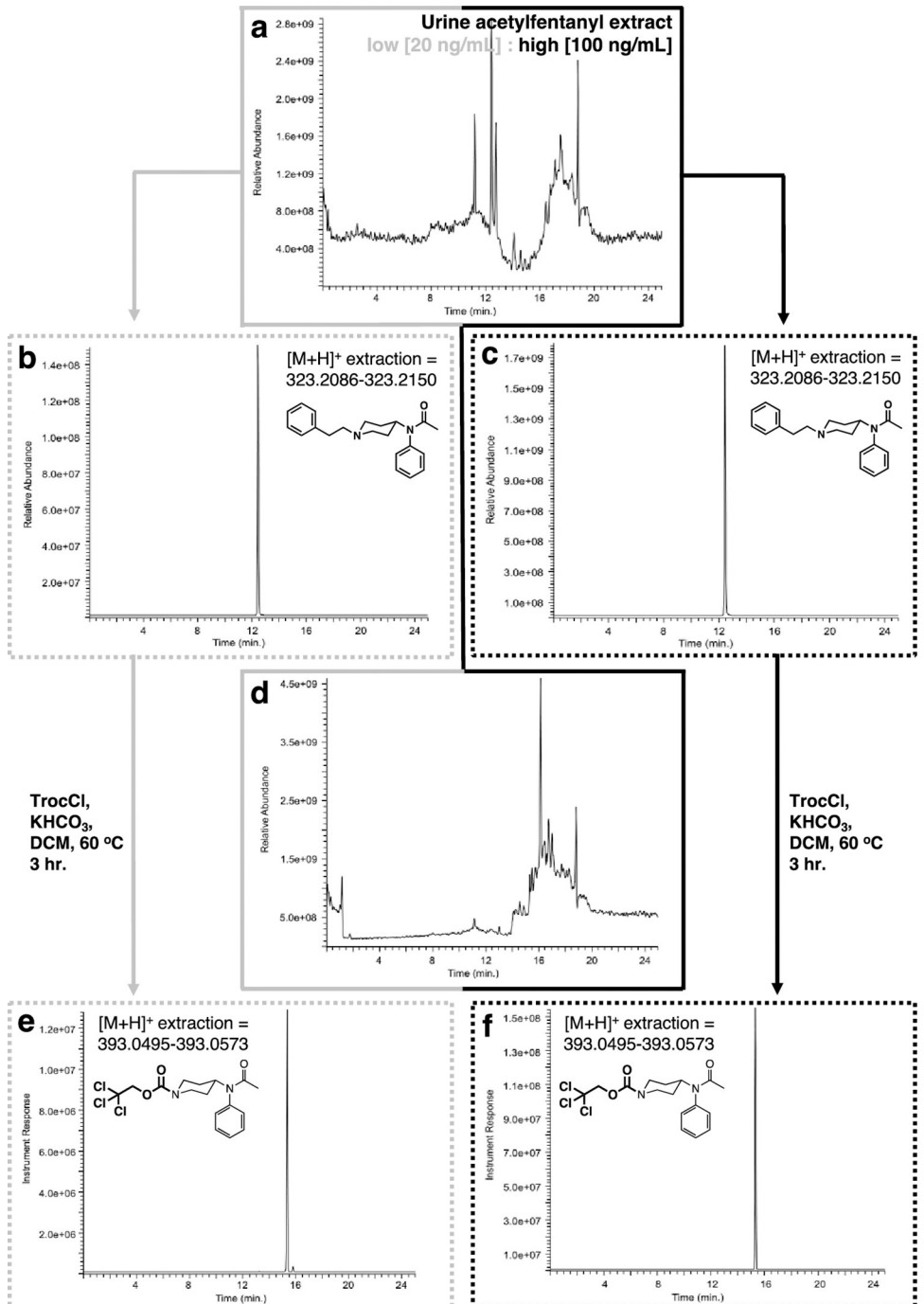

**Fig 14. Analysis by HR-LC-MS of extraction and reaction of acetylfentanyl with Troc-Cl when spiked in urine. (a)**
Total ion chromatogram (TIC) of extracted urine sample; **(b)** extracted ion ($[M+H]^+$ = 323.2118 ± 5 ppm) for
acetylfentanyl (rt = 12.46 min.) for low concentration urine sample (20 ng/mL); **(c)** extracted ion ($[M+H]^+$ =
323.2118 ± 5 ppm) for acetylfentanyl for high concentration urine sample (100 ng/mL); **(d)** Total ion chromatogram
(TIC) of the reaction of extracted urine sample with Troc-Cl; **(e)** extracted ion ($[M+H^+]$ = 393.0534 ± 5 ppm) for Troc-
norfentanyl for low concentration spiked urine sample; **(f)** extracted ion ($[M+H^+]$ = 393.0534 ± 5 ppm) for Troc-
norfentanyl for the high concentration spiked urine sample showing its appearance at rt = 15.42 min.

of the fentanyl-spiked plasma samples (Fig 13). These extracts exhibited complex TICs as appreciated in Fig 13A. Analysis of the TIC traces derived from the low and high concentration samples by high resolution mass extraction for the protonated acetylfentanyl form ($C_{21}H_{27}N_2O^+$ [M+H$^+$ = 323.2118] ± 5 ppm window) yielded clear ion chromatograms for the presence of the opioid (Fig 13B and 13C). These results proved that the extraction part of the protocol was successful as expected and set the stage for the second part of the protocol (reaction with Troc-Cl). Reaction of the residue with Troc-Cl provides a complex mixture of substances as witnessed in the TIC in Fig 13D. High resolution mass extraction of these mixtures for the Troc-noracetylfentanyl product ($C_{16}H_{20}Cl_3N_2O_3^+$ [M+H$^+$ = 393.0534] ± 5 ppm window) led to the direct detection of Troc-norfentanyl for both low and high concentration samples (Fig 13E and 13F). Even though detection of the acetylfentanyl and its Troc-norfentanyl product was found to be straightforward due to their high concentration in the plasma matrix, no clear detection was accomplished for the second product of the reaction, namely 2-phenylethyl chloride. This highlights the importance of using a complementary technique like EI-GC-MS in parallel to get a complete picture of the reaction. In this case HR-LC-MS would seem to be the only technique to employ as its level of detection is orders of magnitude superior to the one featured by EI-GC-MS. However, detection of 2-phenylethyl chloride is an important part of the identification process of the opioid that can be accomplished by EI-GC-MS and not necessarily by HR-LC-MS as noted in this part of the work. A similar set of results were obtained for the acetylfentanyl-spiked urine samples. The urine samples were extracted prior to their modification with Troc-Cl, and then analyzed for intact acetylfentanyl as previously done for the plasma samples. The complex TIC can be appreciated in Fig 14A. Analysis of the TIC traces derived from the low and high concentration samples by high resolution mass extraction for the protonated acetylfentanyl form ($C_{21}H_{27}N_2O^+$ [M+H$^+$ = 323.2118] ± 5 ppm window) yields clear ion chromatograms for the presence of the opioid (Fig 14B and 14C). Reaction of both residues (low and high concentrations) with Troc-Cl provides a complex mixture of substances (Fig 14D) but when analyzed using the high resolution mass extraction for Troc-noracetylfentanyl ($C_{16}H_{20}Cl_3N_2O_3^+$ [M+H$^+$ = 393.0534] ± 5 ppm window) led to the direct detection of this product for both, the low and high concentration-spiked samples (Fig 14E and 14F).

## Conclusion

The reaction between fentanyls and the chloroformate Troc-Cl results in the generation of two predictable products that are unique to the original fentanyl and also abiotic in nature which allows for the identification of the opioid in biological samples. In this work we have demonstrated the successful incorporation of this chemistry in the established identification process of fentanyl and acetylfentanyl in biological matrices present in relevant concentrations in overdose victims using EI-GC-MS and HR-LC-MS. The reaction produces a Troc-modified version of the metabolic products from both opioids, namely norfentanyl and noracetylfentanyl that now provide an additional species to fully confirm the presence of these lethal opioids in these matrices. The procedure consists of the initial extraction of the fentanyls from the matrices followed by treatment of the extracts with Troc-Cl to provide the unique signature products. Both opioids were spiked separately in urine (at 5 and 10 ng/mL) and plasma (at 10 and 20 ng/mL) at two concentrations (one low and one high). Interestingly, even though the extraction step was successful for the low concentration urine sample for fentanyl (~5 ng/mL), the Troc method did not produce any of the desired Troc-norfentanyl highlighting the limitation of the EI-GC-MS part of the protocol when dealing with such low concentrations of the opioid. In contrast, analysis of the high concentration urine sample for fentanyl (~10 ng/mL),

detected the formation of Troc-norfentanyl for its qualitative detection by GC-MS. For the acetylfentanyl spikes, the product arising from the protocol, Troc-noracetylfentanyl was detected in both urine samples (spiked at 20 and 100 ng/mL) as well as the plasma samples (spiked at 50 and 200 ng/mL). Analysis of the same set of samples by HR-LC-MS yielded better results due to the technique's inherently superior sensitivity. The HR-LC-MS method's LOQ (limit of quantitation) for the Troc-norfentanyl and Troc-noracetylfentanyl products was determined to be ~10 ng/mL for both species. Even though the superiority in the detection of these species by HR-LC-MS over EI-GC-MS, the latter method proved to be important in the detection of the second product from the reaction, namely 2-phenylethyl chloride that is crucial in the determination of the original opioid. The work described herein greatly aids in the confirmation of a known fentanyl present in collected urine, plasma and by extension other biological samples amenable to the common extraction procedures described for opioid analysis. However, the method's main strength comes from its ability to react with unknown fentanyls in a predictable manner to yield products that can be not only detected by EI-GC-MS and HR-LC-MS but can then be analyzed to retrospectively identify the original fentanyl which might not be a traditionally used one.

## Supporting information

**S1 File. Supplementary material to the manuscript.**
(PDF)

## Acknowledgments

The authors would like to thank Dr. Carolyn J. Koester and Mr. Armando Alcaraz for engaging discussions regarding GC-MS/LC-MS analysis and their critical reading of the manuscript.

## Author Contributions

**Conceptualization:** Carlos A. Valdez, Todd H. Corzett.

**Data curation:** Carlos A. Valdez, Roald N. Leif.

**Formal analysis:** Roald N. Leif, Todd H. Corzett, Mark L. Dreyer.

**Investigation:** Roald N. Leif, Todd H. Corzett.

**Methodology:** Mark L. Dreyer.

**Writing – original draft:** Carlos A. Valdez.

**Writing – review & editing:** Carlos A. Valdez, Roald N. Leif, Todd H. Corzett, Mark L. Dreyer.

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
