## [Decision Letter · Decision Letter 0]

1 Sep 2022

PONE-D-22-22025Analysis, Identification and  Confirmation of Synthetic Opioids using Chloroformate Chemistry: Retrospective Detection of Fentanyl and Acetylfentanyl in Urine and Plasma Samples by EI-GC-MS and HR-LC-MSPLOS ONE

Dear Dr. Valdez,

Thank you for submitting your manuscript to PLOS ONE. After careful consideration, we feel that it has merit but does not fully meet PLOS ONE’s publication criteria as it currently stands. Therefore, we invite you to submit a revised version of the manuscript that addresses the points raised during the review process.

We look forward to receiving your revised manuscript.

Kind regards,

Joseph Banoub, Ph,D., D. Sc.,FCIC, FRCS

Academic Editor

PLOS ONE

Journal Requirements:

"This document (LLNL-JRNL-XXXX) was prepared as an account of work sponsored by an agency of the United States government. "

"This work was performed under the auspices of the U. S. Department of Energy by Lawrence Livermore National Laboratory under Contract DE-AC52-07NA27344.  The was funded fully by a Mid-Career Research Grant awarded by the Lawrence Livermore National Laboratory (PLS-21-FS-036) to C. A. V.  The funders had no role in study design, data collection and analysis, decision to publish, or preparation of the manuscript. "

Reviewers' comments:

Reviewer's Responses to Questions

**Comments to the Author**

1. Is the manuscript technically sound, and do the data support the conclusions?

Reviewer #1: Yes

Reviewer #2: Yes

2. Has the statistical analysis been performed appropriately and rigorously? 

Reviewer #1: N/A

Reviewer #2: Yes

3. Have the authors made all data underlying the findings in their manuscript fully available?

Reviewer #1: Yes

Reviewer #2: Yes

4. Is the manuscript presented in an intelligible fashion and written in standard English?

Reviewer #1: Yes

Reviewer #2: Yes

5. Review Comments to the Author

Reviewer #1: The study developed in the article submitted for expert appraisal consists of the analysis of the products resulting from the trichloroethoxycarbonylation of fentanyl and acetylfentanyl in urine and plasma matrices. The method described involves the initial extraction of the two synthetic opioids separately from the matrices and then detection of the single product resulting from their reaction with 2,2,2-trichloroethoxycarbonyl chloride, namely Troc-norfentanyl and Troc -noracetylfentanyl. The optimized protocol was evaluated by the authors for its effectiveness in detecting these species formed from fentanyl and acetylfentanyl when the latter is present at low and high levels in urine. The limit of quantification of the method is thus determined at approximately 10 ng/mL for the two species. Although the superiority in the detection of these species by HR-LC-MS, EI-GC-MS proves important in the detection of the second product of the reaction (2-phenylethyl chloride), important in the determination of the opioid. This observation by the authors shows the importance of using analyzes that complement the techniques for analyzing a sample, whether biological or environmental in nature. The present method can thus serve as an additional qualitative confirmation of the presence of a fentanyl in collected urine, plasma and by extension other biological samples which can lend themselves to routine extraction procedures for opioid analysis. The described method can also be used to retrospectively identify unknown fentanyls. It would be interesting to develop this aspect in the conclusion in concrete perspectives.

General comments; The study is perfectly described and the conclusions are supported by results which for some lack a little discussion. In particular a comparative discussion with recently published studies in the field. The conclusion should open on perspectives that need to be developed.

A thorough proofreading should correct minor typographical errors.

Line476 Discussion or conclusion?

line606 line feed

Simplify the titles of the figures and favor putting annotations.

....

Reviewer #2: This is an interesting piece of research of significant topical value. While the use of fentanyl and analogues in a medicinal context is of great benefit, their illicit use is of at least equal (potential) societal damage. The paper is well written and the methodology sound and described in suitable detail. One could perhaps quibble that limits of detection could be more rigorously determined but the primary benefit of the data is not dependent upon their total precision. The figures are well crafted and explained and the deductions sound. The great value of the work is not in just providing another protocol for identifyig and measuring fentanyl but in providing a valuable paradigm for identification not just of the parent compound and currently known derivatives but also for identifying previously unknown novel analogues.

6. PLOS authors have the option to publish the peer review history of their article (what does this mean?). If published, this will include your full peer review and any attached files.

Reviewer #1: No

Reviewer #2: No

---

## [Author Response · Author response to Decision Letter 0]

10 Sep 2022

Dear Reviewers and Editor,

Thank you for reviewing our manuscript titled: “Analysis, Identification and Confirmation of Synthetic Opioids using Chloroformate Chemistry: Retrospective Detection of Fentanyl and Acetylfentanyl in Urine and Plasma Samples by EI-GC-MS and HR-LC-MS”.

We have addressed the first reviewer’s comments and these are highlighted in the “revised manuscript” and present in the “manuscript” new versions of the document. The first reviewers had three comments. The first two were straightforward (i.e., replace discussion by conclusion and a line feed in Line 606). The last one was a suggestion to simplify titles for the figures. We have attempted to reduce the text and insert informative titles for each figure as suggested and per the journal’s request. We hope these changes are sufficient enough so as to not impact the overall description of the figure by the captions below it. Please let us know if we would need to modify further. Again, these changes are highlighted in the “revised manuscript” and present in the “manuscript” new versions of the document.

To the Editor, we have formatted the manuscript per guidelines set by the journal. All these changes can be easily followed by the highlighted “revised manuscript” new version of the document. In addition, we removed the Disclaimer and Auspices statements from the Acknowledgements section. The new, revised funding statement can be found in the "Response to Reviewers" letter.

---

## [Editor Report · Decision Letter 1]

12 Sep 2022

PONE-D-22-22025R1Analysis, Identification and  Confirmation of Synthetic Opioids using Chloroformate Chemistry: Retrospective Detection of Fentanyl and Acetylfentanyl in Urine and Plasma Samples by EI-GC-MS and HR-LC-MSPLOS ONE

Dear Dr. Valdez,

Thank you for submitting your manuscript to PLOS ONE. After careful consideration, we feel that it has merit but does not fully meet PLOS ONE’s publication criteria as it currently stands. Therefore, we invite you to submit a revised version of the manuscript that addresses the points raised during the review process.

We look forward to receiving your revised manuscript.

Kind regards,

Joseph Banoub, Ph,D., D. Sc., FCIC, FRSC

Academic Editor

PLOS ONE

Journal Requirements:

Additional Editor Comments (if provided):

The authors are advised to do the minor corrections suggested by referee 1.
---

## [Author Response · Author response to Decision Letter 1]

13 Sep 2022

Dear Reviewers and Editor,

Thank you for reviewing our manuscript titled: “Analysis, Identification and Confirmation of Synthetic Opioids using Chloroformate Chemistry: Retrospective Detection of Fentanyl and Acetylfentanyl in Urine and Plasma Samples by EI-GC-MS and HR-LC-MS”.

We have addressed the first reviewer’s comments and these are highlighted in the “revised manuscript” and present in the “manuscript” new versions of the document. The second reviewer did not have any specific comments on the manuscript. The first reviewer had three comments. The first two were straightforward (i.e., replace discussion by conclusion and a line feed in Line 606). The last one was a suggestion to simplify titles for the figures. We have attempted to reduce the text and insert informative titles for each figure as suggested and per the journal’s request. Below is an itemized list these changes in the revised version:

1. The word discussion was changed to conclusion in the main text.

2. Line 606 was denoted as a line feed. We have deleted this.

3. Titles of the Figures have been formatted and simplified as much as possible while retaining their clear, descriptive nature. The changes made are itemized below and the edits are highlighted in yellow. A tracked changes version of the Figure file has been also uploaded in the overall review of the manuscript for your convenience.

4. Fig 1. Synthetic opioids belonging to the fentanyl family. (a) Chemical structures of fentanyl, acetylfentanyl and their main metabolites, the potencies of each opioid relative to morphine are given in brackets; (b) commonly employed antidotes against fentanyl poisoning along with their circulation half-lives.

5. Fig 2. Chemical modification of fentanyls. (a) Reaction of fentanyl with 2,2,2-trichloroethoxycarbonyl chloride (Troc-Cl) to yield two unique, abiotic products that bear structural features of the original opioid (grayed boxes); (b) overall protocol to aid in the confirmation of fentanyl and norfentanyl in a biological sample by EI-GC-MS and targeted HR-LC-MS.

6. Fig 3. Extraction and reaction of fentanyl with Troc-Cl when spiked in plasma at two concentrations (10 and 20 ng/mL). (a) GC chromatogram of extracted plasma sample; (b) extracted ion (m/z = 245) chromatogram for fentanyl (rt = 32.8 min.) for low concentration spike; (c) extracted ion (m/z = 245) chromatogram for fentanyl for high concentration spike; (d) GC chromatogram of the reaction between extracted plasma sample and Troc-Cl; (e) extracted ion (m/z = 149) chromatogram for Troc-norfentanyl for the low concentration sample showing no discernible peak while (f) extracted ion (m/z = 149) chromatogram for Troc-norfentanyl for the high concentration sample shows a clear peak at rt = 33.03 min.

7. Fig 4. Mass spectral data associated with the reaction between fentanyl-spiked plasma matrices with Troc-Cl. Mass spectra for (a) Troc-norfentanyl product arising from the treatment of fentanyl-spiked plasma extract (20 ng/mL) with Troc-Cl, note that the spectrum is complex and it is the result of an additional Troc-containing interference (Troc-norfentanyl peaks indicated in red); (b) Expansion of the molecular ion peak region for Troc-norfentanyl where the peaks arising from our product are highlighted in red and those from the Troc-containing interference are in gray; mass spectra for (c) extracted fentanyl when spiked at 10 ng/mL and (d) extracted fentanyl when spiked at 20 ng/mL; (e) mass spectrum for 2-phenylethyl chloride for the lowest fentanyl concentration spike and (f) for the highest fentanyl concentration.

8. Fig 5. Extraction and reaction of fentanyl with Troc-Cl when spiked in urine. (a) GC chromatogram of extracted urine sample; (b) extracted ion (m/z = 245) chromatogram for fentanyl (rt = 32.8 min.) for low concentration urine spike (5 ng/mL); (c) extracted ion (m/z = 245) chromatogram for fentanyl for high concentration urine spike (10 ng/mL); (d) GC chromatogram of the reaction between extracted urine sample and Troc-Cl; (e) extracted ion (m/z = 149) chromatogram for Troc-norfentanyl for the low concentration urine sample (rt = 33.03 min.); (f) extracted ion (m/z = 149) chromatogram for Troc-norfentanyl for the high concentration spiked urine sample.

9. Fig 6. Mass spectral data associated with the reaction between fentanyl-spiked urine matrices with Troc-Cl. Mass spectra for (a) extracted fentanyl when spiked at 5 ng/mL, (b) extracted fentanyl when spiked at 10 ng/mL, (c) Troc-norfentanyl product arising from the treatment of fentanyl-spiked urine extract (10 ng/mL) with Troc-Cl, (d) and 2-phenylethyl chloride, the second confirmatory by-product from the reaction; mass spectra. Mass spectra for Troc-norfentanyl and 2-phenylethyl chloride contain interfering signals from other matrix components, but in both cases the base peaks can be observed (m/z = 149 and m/z = 91 respectively).

10. Fig 7. Extraction and reaction of acetylfentanyl with Troc-Cl when spiked in plasma. (a) GC chromatogram of extracted plasma sample; (b) extracted ion (m/z = 231) chromatogram for acetylfentanyl (rt = 32.3 min.) for low concentration plasma spike (50 ng/mL); (c) extracted ion (m/z = 231) chromatogram for fentanyl for high concentration plasma spike (200 ng/mL); (d) GC chromatogram of the reaction between extracted plasma sample and Troc-Cl; (e) extracted ion (m/z = 135) chromatogram for Troc-norfentanyl for the low concentration plasma sample (rt = 32.6 min.); (f) extracted ion (m/z = 135) chromatogram for Troc-norfentanyl for the high concentration spiked plasma sample.

11. Fig 8. Mass spectral data associated with the reaction between acetylfentanyl-spiked plasma matrices with Troc-Cl. Mass spectra for (a) extracted fentanyl when spiked at 50 ng/mL, (b) Troc-norfentanyl product arising from the treatment of fentanyl-spiked plasma extract (50 ng/mL) with Troc-Cl, (c) and 2-phenylethyl chloride, the second confirmatory by-product from the reaction; mass spectra for (d) extracted fentanyl when spiked at 100 ng/mL, (e) Troc-norfentanyl product arising from the treatment of fentanyl-spiked plasma extract (100 ng/mL) with Troc-Cl, (f) and 2-phenylethyl chloride for the highest concentration spike.

12. Fig 9. Extraction and reaction of acetylfentanyl with Troc-Cl when spiked in urine. (a) GC chromatogram of extracted urine sample; (b) extracted ion (m/z = 231) chromatogram for acetylfentanyl (rt = 32.3 min.) for low concentration urine spike (20 ng/mL); (c) extracted ion (m/z = 231) chromatogram for fentanyl for high concentration urine spike (100 ng/mL); (d) GC chromatogram of the reaction between extracted urine sample and Troc-Cl; (e) extracted ion (m/z = 135) chromatogram for Troc-norfentanyl for the low concentration urine sample (rt = 32.6 min.); (f) extracted ion (m/z = 135) chromatogram for Troc-norfentanyl for the high concentration spiked urine sample.

13. Fig 10. Mass spectral data associated with the reaction between acetylfentanyl-spiked urine matrices with Troc-Cl. Mass spectra for (a) extracted fentanyl when spiked at 20 ng/mL, (b) Troc-norfentanyl product arising from the treatment of fentanyl-spiked urine extract (20 ng/mL) with Troc-Cl, (c) and 2-phenylethyl chloride, the second confirmatory by-product from the reaction; mass spectra for (d) extracted fentanyl when spiked at 100 ng/mL, (e) Troc-norfentanyl product arising from the treatment of fentanyl-spiked plasma extract (100 ng/mL) with Troc-Cl, (f) and 2-phenylethyl chloride for the highest concentration spike.

14. Fig 11. Analysis by HR-LC-MS of extraction and reaction of fentanyl with Troc-Cl when spiked in plasma. (a) Total ion chromatogram (TIC) of extracted plasma sample; (b) extracted ion ([M+H+] = 337.4865 � 5 ppm) for fentanyl (rt = 13.85 min.) for low concentration plasma sample (10 ng/mL); (c) extracted ion ([M+H+] = 337.4865 � 5 ppm) for high concentration in plasma sample (20 ng/mL); (d) Total ion chromatogram (TIC) of reaction between extracted plasma sample with Troc-Cl; (e) extracted ion ([M+H+] = 407.7160 � 5 ppm) for Troc-norfentanyl for low concentration spiked plasma sample; (f) extracted ion ([M+H+] = 407.7160 � 5 ppm) for Troc-norfentanyl for the high concentration spiked plasma sample showing its appearance at rt = 15.91 min.

15. Fig 12. Analysis by HR-LC-MS of extraction and reaction of fentanyl with Troc-Cl when spiked in urine. (a) Ion chromatogram of extracted urine sample; (b) extracted ion ([M+H+] = 337.4865 � 5 ppm) for fentanyl (rt = 13.85 min.) for low concentration urine sample (5 ng/mL); (c) extracted ion ([M+H+] = 337.4865 � 5 ppm) for high concentration in urine sample (10 ng/mL); (d) Ion chromatogram of reaction between extracted urine sample with Troc-Cl; (e) extracted ion ([M+H+] = 407.7160 � 5 ppm) for Troc-norfentanyl for low concentration spiked urine sample; (f) extracted ion ([M+H+] = 407.7160 � 5 ppm) for Troc-norfentanyl for the high concentration spiked urine sample showing its appearance at rt = 15.91 min.

16. Fig 13. Analysis by HR-LC-MS of extraction and reaction of acetylfentanyl with Troc-Cl when spiked in plasma. (a) Total ion chromatogram (TIC) of extracted plasma sample; (b) extracted ion ([M+H]+ = 323.4595 � 5 ppm) for acetylfentanyl (rt = 12.46 min.) for low concentration plasma sample (50 ng/mL); (c) extracted ion ([M+H]+ = 323.4595 � 5 ppm) for acetylfentanyl for high concentration plasma sample (200 ng/mL); (d) Total ion chromatogram (TIC) of the reaction of extracted urine sample with Troc-Cl; (e) extracted ion ([M+H+] = 393.6890 � 5 ppm) for Troc-norfentanyl for low concentration spiked plasma sample; (f) extracted ion ([M+H+] = 393.6890 � 5 ppm) for Troc-norfentanyl for the high concentration spiked plasma sample showing its appearance at rt = 15.42 min.

17. Fig 14. Analysis by HR-LC-MS of extraction and reaction of acetylfentanyl with Troc-Cl when spiked in urine. (a) Total ion chromatogram (TIC) of extracted urine sample; (b) extracted ion ([M+H]+ = 323.4595 � 5 ppm) for acetylfentanyl (rt = 12.46 min.) for low concentration urine sample (20 ng/mL); (c) extracted ion ([M+H]+ = 323.4595 � 5 ppm) for acetylfentanyl for high concentration urine sample (100 ng/mL); (d) Total ion chromatogram (TIC) of the reaction of extracted urine sample with Troc-Cl; (e) extracted ion ([M+H+] = 393.6890 � 5 ppm) for Troc-norfentanyl for low concentration spiked urine sample; (f) extracted ion ([M+H+] = 393.6890 � 5 ppm)for Troc-norfentanyl for the high concentration spiked urine sample showing its appearance at rt = 15.42 min.

We hope these changes are sufficient enough so as to not impact the overall description of the figure by the captions below it. Please let us know if we would need to modify further. Again, these changes are highlighted in the “revised manuscript” and present in the “manuscript” new versions of the document. 

To the Editor, we have formatted the manuscript per guidelines set by the journal. All these changes can be easily followed by the highlighted “revised manuscript” new version of the document. Below is an itemized list of all the changes in the revised version.

1. Line 29: Abstract title changed to font size = 18.

2. Line 57: Introduction title changed to font size = 18.

3. Line 157: Materials and methods title changed to font size = 18.

4. Line 159: Chemicals and reagents title changed to font size = 16.

5. Line 173: EI-GC-MS Analysis Method title changed to font size = 16.

6. Line 191: HR-LC-MS Analysis Method title changed to font size = 16.

7. Line 209: Sample preparation and extraction title changed to font size = 16.

8. Line 231: Results and Discussion title changed to font size = 18.

9. Line 577: Conclusion title (changed from originally Discussion) changed to font size = 18.

10. Line 607: Acknowledgements title changed to font size = 18 and now reads: “The authors would like to thank Dr. Carolyn J. Koester and Mr. Armando Alcaraz for engaging discussions regarding GC-MS/LC-MS analysis and their critical reading of the manuscript.”

11. Lines 613-614: The disclaimer and auspices statements where funding source is mentioned have been removed and added to the bottom of this rebuttal letter.

12. All references annotations within the main text of the manuscript have been changed from parentheses ( ) to brackets [ ].

Line 59: (1, 2) changed to [1, 2]

Line 61: (3) changed to [3]

Line 61: (4, 5) changed to [4, 5]

Line 66: (6) changed to [6]

Line 69: (7-9) changed to [7-9]

Line 71: (10) changed to [10]

Line 73: (11, 12) changed to [11, 12]

Line 76: (13-15) changed to [13-15]

Lines 77-78: (16, 17) changed to [16, 17]

Line 78: (18, 19) changed to [18, 19]

Line 96: (16, 20) changed to [16, 20]

Line 98: (16) changed to [16]

Lines 100-101: (21, 22) changed to [21, 22]

Line 103: (23-26) changed to [23-26]

Line 110: (27, 28) changed to [27, 28]

Line 121: (29-31) changed to [29-31]

Line 125: (32) changed to [32]

Line 130: (32) changed to [32]

Line 140: (33-35) changed to [33-35]

Line 169: (36, 37) changed to [36, 37]

Line 169: (38, 39) changed to [38, 39]

Line 176: (40-43) changed to [40-43]

Line 213: (17) changed to [17]

Line 218: (44) changed to [44]

Line 235: (45) changed to [45]

Line 235: (46) changed to [46]

Line 237: (47) changed to [47]

Line 240: (48, 49) changed to [48, 49]

Line 243: (50, 51) changed to [50, 51]

Line 245: (e.g. drug confiscation) changed to (e.g., drug confiscation)

Line 246: (e.g. hydrochloride) changed to (e.g., hydrochloride)

Line 247: (52, 53) changed to [52, 53]

Line 259: (32) changed to [32]

Line 268: (44) changed to [44]

Line 274: (54) changed to [54]

Line 298: (32) changed to [32]

Line 334: (e.g. benzylfentanyl) changed to (e.g., benzylfentanyl)

Line 403: (55-57) changed to [55-57]

Line 404: (58) changed to [58]

13. All Figure annotations within the main text of the manuscript have been changed from the original (Figure 1) to (Fig 1) and so on. Each and every change is highlighted in yellow in the “revised manuscript” version.

Line 62: (Figure 1a) changed to (Fig 1a)

Line 68: (Figure 1a) changed to (Fig 1a)

Line 76: (Figure 1b) changed to (Fig 1b)

Line 99: (Figure 1a) changed to (Fig 1a)

Line 128: (Figure 2a) changed to (Fig 2a)

Line 135: (Figure 2b) changed to (Fig 2b)

Line 140: (Figure 2b) changed to (Fig 2b)

Line 271: (Figure 3) changed to (Fig 3)

Line 276: (Figure 3a) changed to (Fig 3a)

Line 288: (Figures 3e and 3f) changed to (Figs 3e and 3f)

Line 289: (Figures 3e and 3f) changed to (Figs 3e and 3f)

Line 290: Figures 3e and 3f text changed to Figs 3e and 3f

Line 293: Figure 3f text changed to Fig 3f

Line 296: (Figure 4a) changed to (Fig 4a)

Line 299: Figure 4a text changed to Fig 4a

Line 301: Figure 4b text changed to Fig 4b

Line 304: Figure 4b text changed to Fig 4b

Line 317: Figure 4a text changed to Fig 4a

Line 322: (Figures 4c and 4d) changed to (Figs 4c and 4d)

Line 324: (Figures 4e and 4f) changed to (Figs 4e and 4f)

Line 328: (Figures 4a, 4b and 4f) changed to (Figs 4a, 4b and 4f)

Line 331: (Figure 4e) changed to (Fig 4e)

Line 341: (Figure 5) changed to (Fig 5)

Line 345: (Figure 5a) changed to (Fig 5a)

Line 347: (Figures 5b and 5c) changed to (Fig 5b and 5c)

Line 349: (Figure 5d) changed to (Fig 5d)

Line 350: (Figure 5e) changed to (Fig 5e)

Line 351: (Figure 5f) changed to (Fig 5f)

Line 355: (Figures 4a and 4b) changed to (Fig 4a and 4b)

Line 359: (Figure 6c) changed to (Fig 6c)

Line 371: (Figure 6d) changed to (Fig 6d)

Line 371: Figure 6d in text changed to Fig 6d

Line 375: Figure 6d in text changed to Fig 6d

Line 382: (Figures 6e and 6f) changed to (Fig 6e and 6f)

Line 384: (Figure 6f) changed to (Fig 6f)

Line 408: (Figure 7a) changed to (Fig 7a)

Line 410: (Figures 7b and 7c) changed to (Figs 7b and 7c)

Line 411: (Figure 7d) changed to (Fig 7d)

Line 412: (Figures 7e and 7f) changed to (Figs 7e and 7f)

Line 416: (Figures 8a-f) changed to (Figs 8a-f)

Line 418: (Figures 8b and 8e) changed to (Figs 8b and 8e)

Line 433: (Figure 9a) changed to (Fig 9a)

Lines 435-436: (Figures 9b and 9c) changed to (Figs 9b and 9c)

Line 436: (Figure 9d) changed to (Fig 9d)

Line 437: (Figures 9e and 9f) changed to (Figs 9e and 9f)

Lines 439-440: (Figures 10a and 10d) changed to (Figs 10a and 10d)

Line 443: (Figures 10b-f) changed to (Figs 10b-f)

Line 474: (Figure 11a) changed to (Fig 11a)

Line 479: Figures 11b and 11c in text changed to Figs 11b and 11c

Line 481: Figure 11d in text changed to Fig 11d

Line 486: (Figures 11e and 11f) changed to (Figs 11e and 11f)

Line 503: Figure 12a in text changed to Fig 12a

Lines 506-507: (Figures 12b and 12c) changed to (Figs 12b and 12c)

Line 510: Figure 12d in text changed to Fig 12d

Lines 518-519: (Figures 12e and 12f) changed to (Figs 12e and 12f)

Line 524: (Figures 13-14) changed to (Figs 13-14)

Line 525: (Figure 13) changed to (Fig 13)

Line 526: Figure 13a in text changed to Fig 13a

Line 529: (Figures 13b and 13c) changed to (Figs 13b and 13c)

Line 533: Figure 13d in text changed to Fig 13d

Lines 535-536: (Figures 13e and 13f) changed to (Figs 13e and 13f)

Line 555: Figure 14a in text changed to Fig 14a

Line 558: (Figures 14b and 14c) changed to (Figs 14b and 14c)

Line 560: (Figure 14d) changed to (Fig 14d)

Line 563: (Figures 14e and 14f) changed to (Figs 14e and 14f

14. All References within the main text of the manuscript have been formatted accordingly to fit the journal’s guidelines (including the addition of DOI info). Below is an itemized list of all the references changed and the changes are highlighted.

Lines 645-647: 1. Janssen PAJ, Eddy NB. Compounds Related to Pethidine-IV. New General Chemical Methods of Increasing the Analgesic Activity of Pethidine. J Med Chem. 1960;2(1):31-45. doi: 10.1021/jm50008a003

Lines 649-650: 2. Stanley TH. The History and Development of the Fentanyl Series. J Pain Symptom Manag. 1992;7(3 Suppl):S3–S7. doi: 10.1016/0885-3924(92)90047-L

Lines 652-654: 3. Grass JA. Fentanyl: Clinical Use as Postoperative Analgesic: Epidural/Intrathecal Route. J Pain Symptom Manag. 1992;7(7):419-430. doi: 10.1016/0885-3924(92)90022-a

Lines 656-657: 4. Schug SA, Ting S. Fentanyl Formulations in the Management of Pain: An Update. Drugs. 2017;77(7):747-763. doi: 10.1007/s40265-017-0727-z

Lines 659-660: 5. Chen C, Gupta A. Clinical and pharmacokinetic considerations of novel formulations of fentanyl for breakthrough cancer pain. Pain Manag. 2014;4(5):339-350. doi: 10.2217/pmt.14.32

Lines 663-664: 6. Bhezadi M, Joukar S, Beik A. Opioids and Cardiac Arrhythmia: A Literature Review. Med. Princ. Pract. 2018;27(5):401-414. doi: 10.1159/000492616

Lines 766-767: 7. Socías ME, Wood E. Epidemic of deaths from fentanyl overdose. Brit Med J 2017;358:j4355. DOI: 10.1136/bmj.j4355

Lines 769-771: 8. Gostin LO, Hodge JG, Noe SA. Reframing the Opioid Epidemic as a National Emergency. J. Am. Med. Assoc. 2017;318(16):1539-1540. DOI: 10.1001/jama.2017.13358 

Lines 773-774: 9. Drummer OH. Fatalities Caused by Novel Opioids: A Review. Forensic Sci. Res. 2019;4(2):95-110. DOI: 10.1080/20961790.2018.1460063 

Lines 776-777: 10. Coupland RM. Incapacitating Chemical Weapons: A Year after the Moscow Theatre Siege. Lancet. 2003;362(9393):1346. DOI: 10.1016/S0140-6736(03)14684-3

Lines 779-780: 11. Valdez CA, Leif RN, Mayer BP. An Efficient, Optimized Synthesis of Fentanyl and Related Analogs. PLoS ONE. 2014;9(9):e108250. DOI: 10.1371/journal.pone.0108250

Lines 782-783: 12. Walz AJ, Hsu F-L. An Operationally Simple Synthesis of Fentanyl Citrate. Org. Prep. Proc. Int. 2017;49(5):467-470. DOI: 10.1080/00304948.2017.1374129 

Lines 785-993: 13. Yeung DT, Bough KJ, Harper JR, Platoff GE. National Institutes of Health (NIH) Executive Meeting Summary: Developing Medical Countermeasures to Rescue Opioid-Induced Respiratory Depression (a Trans-Agency Scientific Meeting) - August 6/7, 2019. J. Med. Toxicol. 2020;16(1):87-105. DOI: 10.1007/s13181-019-00750-x 

Lines 995-997: 14. Pardo B, Taylor J, Caulkins J, Reuter P, Kilmer B. The Dawn of a New Synthetic Opioid Era: The Need for Innovative Interventions. Addiction. 2021;116(6):1304-1312. DOI: 10.1111/add.15222

Lines 999-1001: 15. France CP, Ahern G, Averick S, Enright HA, Esmaeli-Azad B, Federico A, et al. Countermeasures for Preventing and Treating Opioid Overdose. Clin. Pharmacol. Ther. 2020;109(3):578-590. DOI: 10.1002/cpt.2098

Lines 1003-1006: 16. Valdez CA, Leif RN, Hok S, Hart BR. Analysis of Chemical Warfare Agents by Gas Chromatography-Mass Spectrometry: Methods for Their Direct Detection and Derivatization Approaches for the Analysis of Their Degradation Products. Rev. Anal. Chem. 2018;37(1):1-26. DOI: 10.1515/revac-2017-0007. 

Lines 1008-1010: 17. Valdez CA. Gas Chromatography-Mass Spectrometry Analysis of Synthetic Opioids Belonging to the Fentanyl Class: A Review. Crit. Rev. Anal. Chem. 2021. DOI:10.1080/10408347.2021.1927668.

Lines 1204-1206: 18. Kumar K, Morgan DJ, Crankshaw DP. Determination of Fentanyl and Alfentanil in Plasma by High-Performance Liquid Chromatography with Ultraviolet Detection. J. Chromatogr. 1987;419:464-468. DOI: 10.1016/0378-4347(87)80317-1

Lines 1208-1210: 19. Cooreman S, Deprez C, Martens F, Van Bocxlaer J, Croes K. A Comprehensive LC-MS-Based Quantitative Analysis of Fentanyl-Like Drugs in Plasma and Urine. J. Sep. Sci. 2010;33(17-18):2654-2662. DOI: 10.1002/jssc.201000330

Lines 1212 -1214: 20. Bévalot F, Cartiser N, Bottinelli C, Fanton L, Guitton, J. Vitreous humor analysis for the detection of xenobiotics in forensic toxicology: a review. Forensic Toxicol. 2016;34, 12-40. DOI: 10.1007/s11419-015-0294-5 

Lines 1216-1218: 21. Feierman DE, Lasker JM. Metabolism of fentanyl, a synthetic opioid analgesic, by human liver microsomes. Role of CYP3A4. Drug Metabolism and Disposition, 1996;24(9):932-939. 

Lines 1220-1223: 22. Steuer AE, Williner E, Staeheli SN, Kraemer T. Studies on the metabolism of the fentanyl-derived designer drug butyrfentanyl in human in vitro liver preparations and authentic human samples using liquid chromatography-high resolution mass spectrometry (LC-HRMS). Drug Test. Anal. 2017;9(7),1085-1092. DOI: 10.1002/dta.2111 

Lines 1409-1411: 23. Day J, Slawson M, Lugo RA, Wilkins D. Analysis of Fentanyl and Norfentanyl in Human Plasma by Liquid Chromatography-Tandem Mass Spectrometry Using Electrospray Ionization. J. Anal. Toxicol. 2003;27(7):513-516. DOI: 10.1093/jat/27.7.513

Lines 1413-1415: 24. Verplaetse R, Henion J. Quantitative determination of opioids in whole blood using fully automated dried blood spot desorption coupled to on-line SPE-LC-MS/MS. Drug Test. Anal. 2016;8(1), 30-38. DOI: 10.1002/dta.1927

Lines 1417-1420: 25. Strayer KE, Antonides HM, Juhascik MP, Daniulaityte R, Sizemore IE. LC-MS/MS-Based Method for the Multiplex Detection of 24 Fentanyl Analogues and Metabolites in Whole Blood at Sub ng�mL–1 Concentrations. ACS Omega. 2018;3(1):514-523. DOI: 10.1021/acsomega.7b01536

Lines 1422-1424: 26. Qin N, Shen M, Xiang P, Wen D, Shen B, Den H, et al. Determination of 37 fentanyl analogues and novel synthetic opioids in hair by UHPLC-MS/MS and its application to authentic cases. Sci. Rep. 2020;10(1):11569. DOI: 10.1038/s41598-020-68348-w

Lines 1426-1428: 27. Mayer BP, Valdez CA, DeHope AJ, Spackman PE, Williams, AM. Statistical Analysis of the Chemical Attribution Signatures of 3-Methylfentanyl and Its Methods of Production. Talanta. 2018;186:645-654. DOI: 10.1016/j.talanta.2018.02.026

Lines 1629-1632: 28. Ovenden SPB, McDowall LJ, McKeown HE, McGill NW, Jones OAH, Pearson JR, et al. Investigating the Chemical Impurity Profiles of Fentanyl Preparations and Precursors to Identify Chemical Attribution Signatures for Synthetic Method Attribution. Forensic Sci. Int. 2021;321:110742. DOI: 10.1016/j.forsciint.2021.110742

Lines 1634-1635: 29. Stein S. Mass spectral reference libraries: An ever-expanding resource for chemical identification. Anal. Chem. 2012;84(17): 7274-7282. DOI: 10.1021/ac301205z 

Lines 1637-1639: 30. Nyanyira C. The OPCW Central Analytical Database, Chemical Weapons Convention Chemicals Analysis: Sample Collection, Preparation and Analytical Methods (Ed: M. Mesilaakso). John Wiley, Chichester, 2005, p. 133-149.

Lines 1641-1644: 31. Valdez CA, Leif RN, Hok S, Alcaraz A. Assessing the reliability of the NIST library during routine GC-MS analyses: Structure and spectral data corroboration for 5,5-diphenyl-1,3-dioxolan-4-one during a recent OPCW proficiency test. J. Mass Spectrom. 2018;53(5):419-422. DOI: 10.1002/jms.4073

Lines 1646-1649: 32. Valdez CA, Leif RN, Sanner RD, Corzett TH, Dreyer ML, Mason KE. Structural modification of fentanyls for their retrospective identification by gas chromatographic analysis using chloroformate chemistry. Sci. Rep. 2021;11(1):22489. DOI: 10.1038/s41598-021-01896-x

Lines 1844-1846: 33. McIntyre IM, Gary RD, Estrada J, Nelson CL. Antemortem and Postmortem Fentanyl Concentrations: A Case Report. Int. J. Legal Med. 2014;128(1): 65-67. DOI: 10.1007/s00414-013-0897-5

Lines 1848-1849: 34. Cunningham SM, Haikal NA, Kraner JC. Fatal Intoxication with Acetyl Fentanyl. J. Forensic Sci. 2016;61(Suppl 1):S276-S280. DOI: 10.1111/1556-4029.12953 

Lines 1851-1853: 35. Yonemitsu K, Sasao A, Mishima S, Ohtsu Y, Nishitani Y. A Fatal Poisoning Case by Intravenous Injection of "Bath Salts" Containing Acetyl Fentanyl and 4-Methoxy PV8. Forensic Sci. Int. 2016;267:e6-e9. DOI: 10.1016/j.forsciint.2016.08.025

Lines 1855-1858: 36. Wnuk SF, Valdez CA, Khan J, Moutinho P, Robins MJ, Yang X, et al. Doubly homologated dihalovinyl and acetylene analogues of adenosine: Synthesis, interaction with S-adenosyl-L-homocysteine hydrolase, and antiviral and cytostatic effects. J. Med. Chem. 2000;43(6):1180-1186. DOI: 10.1021/jm990486y 

Lines 1860-1863: 37. Wnuk SF, Ro B-O, Valdez CA, Lewandowska E, Valdez NX, Sacasa PR, et al. Sugar-modified conjugated diene analogues of adenosine and uridine: Synthesis, interaction with S-adenosyl-L-homocysteine hydrolase, and antiviral and cytostatic effects. J. Med. Chem. 2002;45:2651-2658. DOI: 10.1021/jm020064f

Lines 2100-2103: 38. Showalter BM, Reynolds MM, Valdez CA, Saavedra JE, Davies KM, Klose JR, et al. Diazeniumdiolate ions as leaving groups in anomeric displacement reactions: a protection-deprotection strategy for ionic diazeniumdiolates. J. Am. Chem. Soc. 2005;127(41):14188-14189. DOI: 10.1021/ja054510a

Lines 2105-2108: 39. Valdez CA, Saavedra JE, Showalter BM, Davies KM, Wilde TC, Citro ML, et al. Hydrolytic reactivity trends among potential prodrugs of the O2-glycosylated diazeniumdiolate family. Targeting nitric oxide to macrophages for antileishmanial activity. J. Med. Chem. 2008;51(13):3961-3970. DOI: 10.1021/jm8000482

Lines 2110-2114: 40. Valdez CA, Leif RN, Hok S, Vu AK, Salazar EP, Alcaraz A. Methylation protocol for the retrospective detection of isopropyl-, pinacolyl- and cyclohexylmethylphosphonic acids, indicative markers for the nerve agents sarin, soman and cyclosarin, at low levels in soils using EI-GC-MS. Sci. Total Environ. 2019;683:175-184. DOI: 10.1016/j.scitotenv.2019.05.205 

Lines 2116-2119: 41. Valdez CA, Leif RN, Alcaraz A. Effective methylation and identification of phosphonic acids relevant to chemical warfare agents mediated by trimethyloxonium tetrafluoroborate for their qualitative detection by gas chromatography-mass spectrometry. Anal. Chim. Acta, 2016;933:134-143. DOI: 10.1016/j.aca.2016.05.034

Lines 2350-2354: 42. Valdez CA, Marchioretto MK, Leif RN, Hok, S. Efficient derivatization of methylphosphonic and aminoethylsulfonic acids related to nerve agents simultaneously in soils using trimethyloxonium tetrafluoroborate for their enhanced, qualitative detection and identification by EI-GC–MS and GC–FPD. Forensic Sci. Int. 2018;288:159-168. DOI: 10.1016/j.forsciint.2018.04.041

Lines 2356-2359: 43. Valdez CA, Leif RN, Hok, S. Carbene-based difluoromethylation of bisphenols: Application to the instantaneous tagging of bisphenol A in spiked soil for its detection and identification by electron ionization gas chromatography-mass spectrometry. Sci. Rep. 2019;9:17360. DOI: 10.1038/s41598-019-53735-9 

Lines 2361-2363: 44. Foerster EH, Mason MF. Preliminary studies on the use of n-butylchloride as an extractant in a drug screening procedure. J. Forensic Sci. 1974;19(1):155-162. PMID: 4853734

Lines 2365-2367: 45. Strano-Rossi S, Alvarez I, Tabernero MJ, Cabarcos P, Fernández P, Bermejo, AM. Determination of Fentanyl, Metabolite and Analogs in Urine by GC/MS. J. Applied Toxicol. 2011;31(7):649–654. DOI: 10.1002/jat.1613

Lines 2542-2543: 46. Gardner MA, Sampsel S, Jenkins WW, Owens JE. Analysis of Fentanyl in Urine by DLLME-GC-MS. J. Anal. Toxicol. 2015;39(2):118-125. DOI: 10.1093/jat/bku136

Lines 2545-2548: 47. Strano-Rossi S, Bermejo AM, de la Torre X, Botrè F. Fast GC-MS Method for the Simultaneous Screening of THCCOOH, Cocaine, Opiates and Analogues Including Buprenorphine and Fentanyl, and Their Metabolites in Urine. Anal. Bioanal. Chem. 2011;399(4):1623-1630. DOI: 10.1007/s00216-010-4471-4

Lines 2550-2552: 48. Sisco E, Burns A, Moorthy AS. Development and evaluation of a synthetic opioid targeted gas chromatography mass spectrometry (GC-MS) method. J. Forensic Sci. 2021;66:2369-2380. DOI: 10.1111/1556-4029.14877 

Lines 2554-2556: 49. Moore A, Foss J, Juhascik M, Botch-Jones S, Kero F. Rapid Screening of opioids in seized street drugs using ambient ionization high resolution time-of-flight mass spectrometry. Forensic Chem. 2019;13:100149. DOI: 10.1016/j.forc.2019.100149

Lines 2558-2560: 50. Weldon ST, Perry DF, Cork RC, Gandolfi AJ. Detection of Picogram Levels of Sufentanil by Capillary Gas Chromatography. Anesthesiology. 1985;63:684-687. DOI: 10.1097/00000542-198512000-00021

Lines 2769-2773: 51. Van Nimmen NFJ, Poels KLC, Veulemans HAF. Highly Sensitive Gas Chromatographic-Mass Spectrometric Screening Method for the Determination of Picogram Levels of Fentanyl, Sufentanil and Alfentanil and Their Major Metabolites in Urine of Opioid Exposed Workers. J. Chromatogr. B Analyt. Technol. Biomed. Life Sci. 2004;804(2):375-387. DOI: 10.1016/j.jchromb.2004.01.044

Lines 2775-2777: 52. Gerace E, Salomone A, Vincenti M. Analytical approaches in fatal intoxication cases involving new synthetic opioids. Curr. Pharm. Biotechnology, 2018;19, 113-123. DOI: 10.2174/1389201019666180405162734

Lines 2779-2781: 53. UNODC. Recommended Methods for the Identification and Analysis of Fentanyl and Its Analogues in Biological Specimens. United Nations Office on Drugs and Crime, Vienna, 2017. DOI: 10.18356/aca7aca5-en.

Lines 2783-2785: 54. Anderson DT, Muto JJ. Duragesic Transdermal Patch: postmortem Tissue Distribution of Fentanyl in 25 Cases. J. Anal. Toxicol. 2000;24:627-634. DOI: 10.1093/jat/24.7.627

Lines 2787-2789: 55. Ogilvie L, Stanley C, Lewis L, Boyd M, Lozier M. Notes from the Field: Acetyl Fentanyl Overdose fatalities-Rhode Island, March-May 2013. Morbidity and Mortality Weekly Report, 2013;62:703-704.

Lines 2960-2962: 56. McIntyre IM, Trochta A, Gary RD, Malamatos M, Lucas JR. An Acute Acetyl Fentanyl Fatality: A Case Report with Postmortem Concentrations. J. Anal. Toxicol. 2015;39(6):490-494. DOI: 10.1093/jat/bkv043

Lines 2964-2965: 57. Fort C, Curtis B, Nichols C, Niblo, C. Acetyl Fentanyl Toxicity: Two Case Reports. J. Anal. Toxicol. 2016;40(9):754-757. DOI: 10.1093/jat/bkw068 

Lines 2967-2969: 58. Katselou M, Papoutsis I, Nikolaou P, Spiliopoulou C, Athanaselis S. Old Opioids, New Concerns: The Case of Acetylfentanyl. Forensic Toxicol. 2016;34:201-212. DOI: 10.1007/s11419-016-0310-4 

In addition, we removed the Disclaimer and Auspices statements from the Acknowledgements section. The new, revised funding statements are below.

“Disclaimer: This document (LLNL-JRNL-839586) was prepared as an account of work sponsored by an agency of the United States government. Neither the United States government nor Lawrence Livermore National Security, LLC, nor any of their employees makes any warranty, expressed or implied, or assumes any legal liability or responsibility for the accuracy, completeness, or usefulness of any information, apparatus, product, or process disclosed, or represents that its use would not infringe privately owned rights. Reference herein to any specific commercial product, process, or service by trade name, trademark, manufacturer, or otherwise does not necessarily constitute or imply its endorsement, recommendation, or favoring by the United States government or Lawrence Livermore National Security, LLC. The views and opinions of authors expressed herein do not necessarily state or reflect those of the United States government or Lawrence Livermore National Security, LLC, and shall not be used for advertising or product endorsement purposes. 

Auspices statement: This work was performed under the auspices of the U. S. Department of Energy by Lawrence Livermore National Laboratory under Contract DE-AC52-07NA27344. The was funded fully by a Mid-Career Research Grant awarded by the Lawrence Livermore National Laboratory (PLS-21-FS-036) to C. A. V.”

Please let us know if there are other edits/corrections that need to be amended. We might have missed some and we apologize in advance for this.

Best regards,

Carlos Valdez and co-authors

---

## [Editor Report · Decision Letter 2]

27 Sep 2022

Analysis, Identification and  Confirmation of Synthetic Opioids using Chloroformate Chemistry: Retrospective Detection of Fentanyl and Acetylfentanyl in Urine and Plasma Samples by EI-GC-MS and HR-LC-MS

PONE-D-22-22025R2

Dear Dr. Valdez,

We’re pleased to inform you that your manuscript has been judged scientifically suitable for publication and will be formally accepted for publication once it meets all outstanding technical requirements.

Kind regards,

Joseph Banoub, Ph,D., D. Sc.

Academic Editor

PLOS ONE
---

## [Editor Report · Acceptance letter]

7 Oct 2022

PONE-D-22-22025R2 

Analysis, Identification and  Confirmation of Synthetic Opioids using Chloroformate Chemistry: Retrospective Detection of Fentanyl and Acetylfentanyl in Urine and Plasma Samples by EI-GC-MS and HR-LC-MS 

Dear Dr. Valdez:

I'm pleased to inform you that your manuscript has been deemed suitable for publication in PLOS ONE. Congratulations! Your manuscript is now with our production department. 

Kind regards, 

on behalf of

Dr. Joseph Banoub 

Academic Editor

PLOS ONE